# A Modelling Approach for the Assessment of Wave-Currents Interaction in the Black Sea

Salvatore Causio [1,*] , Stefania A. Ciliberti [1] , Emanuela Clementi [2], Giovanni Coppini [1] and Piero Lionello [3]

1   Fondazione Centro Euro-Mediterraneo sui Cambiamenti Climatici, Ocean Predictions and Applications Division, 73100 Lecce, Italy; stefania.ciliberti@cmcc.it (S.A.C.); giovanni.coppini@cmcc.it (G.C.)
2   Fondazione Centro Euro-Mediterraneo sui Cambiamenti Climatici, Ocean Modelling and Data Assimilation Division, 40127 Bologna, Italy; Emanuela.clementi@cmcc.it
3   Department of Biological and Environmental Sciences and Technologies, University of Salento—DiSTeBA, 73100 Lecce, Italy; piero.lionello@unisalento.it
*   Correspondence: salvatore.causio@cmcc.it

**Abstract:** In this study, we investigate wave-currents interaction for the first time in the Black Sea, implementing a coupled numerical system based on the ocean circulation model NEMO v4.0 and the third-generation wave model WaveWatchIII v5.16. The scope is to evaluate how the waves impact the surface ocean dynamics, through assessment of temperature, salinity and surface currents. We provide also some evidence on the way currents may impact on sea-state. The physical processes considered here are Stokes–Coriolis force, sea-state dependent momentum flux, wave-induced vertical mixing, Doppler shift effect, and stability parameter for computation of effective wind speed. The numerical system is implemented for the Black Sea basin (the Azov Sea is not included) at a horizontal resolution of about 3 km and at 31 vertical levels for the hydrodynamics. Wave spectrum has been discretised into 30 frequencies and 24 directional bins. Extensive validation was conducted using in-situ and satellite observations over a five-year period (2015–2019). The largest positive impact of wave-currents interaction is found during Winter while the smallest is in Summer. In the uppermost 200 m of the Black Sea, the average reductions of temperature and salinity error are about −3% and −6%, respectively. Regarding waves, the coupling enhanced the model skill, reducing the simulation error, about −2%.

**Keywords:** Black Sea; wave-current interaction; NEMOv4; WaveWatchIII

## 1. Introduction

Wave–currents interaction has recently gained interest in the field of coastal and ocean forecasting [1,2]. Wind-wave-current processes control the momentum and energy exchange between the atmosphere and the ocean and must be better understood and resolved. The physics of wave-currents interaction depend on the kinematics and dynamics of the wave field. These include processes such as wind-wave growth, nonlinear wave-wave interactions, wave-currents interaction and wave dissipation, all of which can only be accurately represented using wave models. Reducing the uncertainties in the nowadays forecasting models that result from the non-linear feedback between the currents [3], water level variations [4,5] and wind waves [6] is therefore essential. Over the past 40 years, a considerable amount of research has been devoted to investigating the role of ocean waves in the air-sea interaction. An accurate representation of ocean surface waves has been recognised as essential in various applications, ranging from marine meteorology to ocean and coastal engineering to operational forecasting [7,8].

Important contributions to the field of wave-currents interaction include the pioneering work of [9] on what is now referred to as the Stokes–Coriolis force, the work of [10] on wave-driven ocean circulation and of [11] on upper ocean mixing by wave breaking. The impact of the oceanic wave field on upper-ocean mixing and mean properties has

been examined using various model experiments [11–17]. Most find that waves appear to have a profound impact on the upper part of the ocean, but there is still considerable disagreement about which processes are more important. Waves define the mixing in the oceanic surface boundary layer (OSBL) via breaking and Langmuir turbulence. For example, ref. [18] identified Langmuir turbulence over wide areas of the global ocean and particularly in the Southern Ocean. In this region, they showed that the inclusion of the effect of surface waves on upper-ocean mixing during summertime led to a reduction in systematic bias in the OSBL depth. Jansen et al. 2013 [17] showed the positive impact of wave breaking on the daily cycle of the sea surface temperature. Babanin et al. 2012 [19] investigated the mixing process related to the orbital motion of non-breaking waves.

All of these wave-driven processes influence the vertical structure of the temperature and current fields in the mixed layer in general, and in the upper few meters in particular. This has implications for coupled models as these processes will affect the feedback between the ocean and the atmosphere [17]. However, on shorter time scales and at higher spatial resolution these processes will influence the drift of objects and pollutants on the sea surface or that are partially or wholly submerged. This has practical implications for oil spill modelling [20] and search and rescue [21–23]. Studies focusing on the Baltic Sea [24] and the North Sea [25] apply the coupling approach previously implemented by [26,27], which consists of coupling the wave model with the circulation model. Staneva et al. 2021 [28] in particular demonstrated how the Eulerian currents provided by a coupled NEMO-WAM model in the North Sea can improve the representation of particle transports, dealing with a number of possible impacts on downstream services for marine litter, oil spill and environmental water properties. This was then adapted to include the Stokes–Coriolis effect, the sea-state-dependent momentum and energy fluxes. These studies show that coupling the models has a pronounced effect on vertical temperature distribution and mesoscale events, and improves predictions of sea level and currents during storm events. In the Mediterranean Sea, Ref. [29] presents an approach based on the assumption that the currents are driven by surface wind stresses that in turn are a function of the sea state, while the sea state depends on the wind speed and the currents. These complex feedback mechanisms can be modelled by coupling hydrodynamic and wave models, which to date have been developed separately. Examples are provided by [8,30] for the North-West European Shelf system. In particular by understanding the role of ocean-wave coupling in an eddy-resolving operational configuration to improve the predictability of sea surface temperature, surface and sea bed salinity, sea surface height and currents.

The active literature production on this topic and the interest in regional modelling in particular for marginal seas induced our attention in understanding the complex mechanisms of wave-currents interaction in the Black Sea. This basin acts as a small-scale laboratory for investigating processes that are common to numerous areas of the world's oceans. Considering that the water and salt balances can be controlled, and the scales are smaller than in the global ocean, the basin is a useful test region for developing models, which can then be applied at larger scales [31].

From the physical point of view, the Black Sea is an inland sea, one of the largest and deepest basins in the world and the farthest to the east among the seas of the Atlantic Ocean. It works with a complex straits system—the Kerch Strait connects the Black Sea with the Azov Sea; the Bosphorus Strait makes the connection with the Marmara Sea which is connected in turn to the Mediterranean Sea through the Dardanelles Strait [32]. Straits system and limited water exchange with the open ocean play an important role in the water balance, influencing the ventilation, stratification and water mass formation [31,33]. Rivers are one of the strongest contributors of freshwater with impact to the general circulation in the Black Sea together with wind stress and bathymetric peculiarities: in particular, the North-West shelf region is dominated by the Danube River floodplain and outflow. The Danube, together with the Dniepr and the Dniester and the dense distribution of rivers along the overall coastline contribute to a positive water budget ($\sim 5.4 \times 10^5$ km$^3$), making the Black Sea a typical estuarine basin [34–36]. The Black Sea

circulation is structured usually in two connected gyre systems encompassing the basin (the Rim Current) and quasi-stationary anticyclonic eddies in the coastal zone, such as Batumi, Sevastopol, Caucasian, Sakarya, Sinop ones [30,37,38]. Some scientific works focused attention on reconstructing the past ocean state in the Black Sea [39] with reference to mesoscale circulation and water mass properties [40–42]. The wind is the main driving force in creating a cyclonic general circulation. Simulation [43] and observations [44] showed that there are three major regions with different regimes of currents: (a) coastal zone of very variable flow, with currents speeds of up to 20–30 cm/s; (b) the main Black Sea currents zone, which has jet-stream character, with a width of 40–80 km and speed of 40–50 cm/s, reaching values of 1–1.5 m/s; (c) the open sea area, where the velocity of the current decreases gradually from the periphery to the centre, not exceeding 5–15 cm/s.

Another important feature of the Black Sea, strictly related to convective processes due to the thermohaline inertia in the upper layer and impact of air-sea fluxes, is the Cold Intermediate Layer (CIL) [45]. CIL is a sub-surface water mass that results from the Winter convective mixing in the centres of cyclonic gyres and shelf areas [32]. CIL formation in the Black Sea depends on the fact that Winter convection is limited by the shallow depth of halocline (such as in the Baltic Sea, the Sea of Okhotsk, the Gulf of St. Lawrence). A consequence of the vertical stratification is that the surface layer (about 0 to 50 m) is well oxygenated while the deep layer (100 m to 2000 m) is anoxic and contains high sulphide concentrations [46].

The wind regime over the Black Sea results from the cyclonic and anticyclonic activity over Europe. The Island cyclone and the Azores anticyclone acts throughout the year, while the Mediterranean cyclones and Siberian anticyclone are more effective in Winter [47]. The Black Sea is dominated by Northerly winds in the West and the North of the sea; Eastern and South-Easterly winds are typical for the East and South-East of the sea. In Spring and Summer, in the Western part of the sea, under the influence of the Azores High, the frequency of Western, South-Western and Southern winds increases [48–51]. South-eastern and Southern coasts of the sea are characterised by weak winds (average annual wind speed < 3 m/s); in the Western and Northwestern parts of the sea, as well as in the Kerch Strait area, stronger winds are observed (average annual wind speed > 4 m/s, at some stations > 5 m/s) [32].

Wind waves in the Black Sea have been widely studied by many researchers from the surrounding regions [52–56]. Brief information about Black Sea wave average climate based on long simulation (1949–2010) is here provided from one of the more recent works [57]. They showed that the average wave parameters are at a maximum in Winter, with maximal values of average Hs exceeding 0.95 m. Areas of most expressed storminess in terms of all average values correspond to the central part of the basin. The same spatial pattern is also observed in the distribution of Springtime averaged wave parameters, but their values are significantly lower (maximal average Hs 0.65 m). The calmest wave field corresponds to Summertime, where Hs > 0.5 m occurred only in small areas of the basin.

From a technical point of view, there are two ways to investigate the impact of ocean-wave interactions: offline or online. Offline coupling e.g., [27,58] implies that the model which provides the fields (sender) runs independently and before the receiving model (receiver). The output from the sender is subsequently used, as the other forcing, by the receiver. This methodology has computational and timing benefits but does not consider the mutual interaction between the two components of the forced system. In the online coupling e.g., [7,28] both the cores are integrated simultaneously, allowing feedback at a predefined exchange time. This methodology more intimately reflects the wave-currents interaction, but it requires an external coupler to drive the models' communication, and the overall running time is almost the sum of both the models running times.

The aim of this work is the assessment, for the first time, of the effects due to the reciprocal interaction between hydrodynamics and waves in the Black Sea through a forced system based on NEMO v4.0 for the hydrodynamics and Wave Watch III for the waves. The main scope is then to improve the representation of the physical processes and dynamics

in the Black Sea region by accounting for the role of waves. To complete the overall investigation, this work presents also some numerical results on the impact of improved hydrodynamics on the sea-state.

The coupling strategy here proposed is based on the exchange of surface Stokes drift, wave energy to the ocean, sea-state dependent momentum flux, currents and the air-sea temperature difference, with the main purpose to improve firstly the mixing process in the hydrodynamical core model. The effects have been evaluated via comparison with a standalone run (without forcing) for both waves and hydrodynamics.

This paper is organised as follows. In Section 2 we present the model system and the coupling strategy. Section 3 illustrates the numerical experiment and validation strategy. Section 4 gives the numerical model results and a comparison with observations, and Section 5 provides conclusions and future works.

## 2. The Modelling System

The numerical system presented in this work is a reciprocal forced hydrodynamic circulation model with a third-generation spectral wind-wave model, implemented in the Black Sea basin. This section presents information about the horizontal numerical grid, which is common to the implementation of both wave and hydrodynamical models. The specific features of this grid, such as spectral discretization for waves and vertical discretization for hydrodynamics, are detailed in the related subsections.

### 2.1. Atmospheric Forcing, Numerical Grid and Bathymetry

Both hydrodynamics and waves are forced by the high-resolution atmospheric model at a horizontal resolution of $0.125° \times 0.125°$ and 6-h frequency, as produced by the European Centre for Medium-range Weather Forecasts (ECMWF, [59]). The system uses Zonal and Meridional Components of the 10 m wind (m/s), Total Cloud Cover (%), 2 m Air Temperature (K), 2 m Dew Point Temperature (K) and mean sea-level pressure (Pa). Precipitation ($Kg/m^2/s$) data are provided as monthly climatologies from GPCP [60,61].

The Black Sea region has been discretised in horizontal space using a regular mesh of $395 \times 215$ grid points, at $0.037°$ (longitude) $\times 0.0278°$ (latitude) spatial resolution in spherical coordinates (Figure 1).

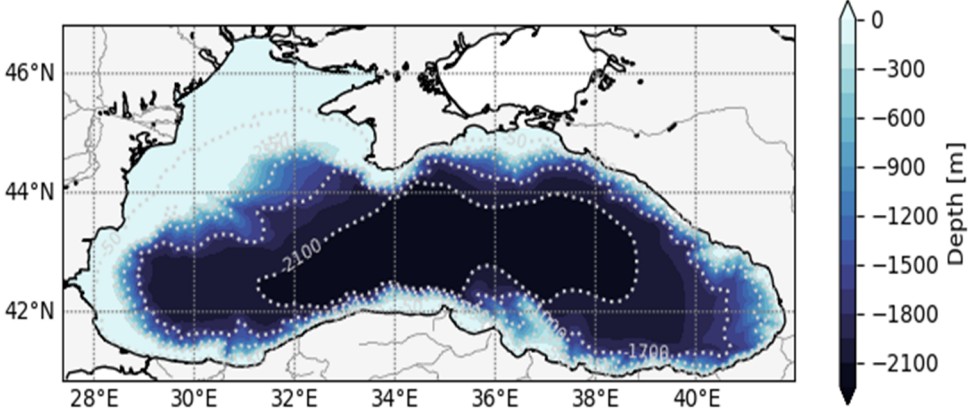

**Figure 1.** The Black Sea domain and bathymetry.

The bathymetric dataset used in this work is the GEBCO_14 [62] at 30″ grid resolution. The GEBCO_14 in the Black Sea basin was merged with a high-resolution dataset for the Bosphorus Strait, as described in [63].

### 2.2. Hydrodynamical Model

The hydrodynamical core of the forced system is based on the Nucleus for European Modelling of the Ocean (NEMO, version v4.0 [64]). The NEMO code solves primitive equations (derived through assuming hydrostatic and incompressible approximations),

and we used the split-explicit free surface formulation. The vertical discretization consists of 31 unevenly spaced z-levels with partial step, with a stretching factor equal to 5, the maximum stretching at the 25th level and a minimum thickness equal to 5 m. A time step of 150 s is used, together with a barotropic time step equal to 1.5 s. The model was initialised in January 2014 using 3D temperature and salinity climatological fields, as provided in the framework of SeaDataCloud v1 for the Black Sea basin [65]. The model computes air-sea fluxes using bulk formulation as implemented for the Mediterranean Sea [66], designed to handle the ECMWF atmospheric forcing data for computing momentum, heat and water fluxes [67,68].

The model considers 72 rivers, whose discharge is estimated from monthly climatological datasets developed in the framework of the SESAME project [69]. The major rivers (e.g., the Danube, Dniepr, Dniester) have been represented as multiple distributed sources and the remainder as source points. The Danube river is represented accounting for the three main branches—the Chilia, the Sulina and the St. George (Figure 2a). Its discharge distribution is set according to [70], with the Chilia accounting for 52%, Sulina for 20% and the St. George 28%. The impact of the Bosphorus Strait on the Black Sea dynamics is accounted for in terms of a surface boundary condition, considering the barotropic transport, which has been computed to balance the freshwater fluxes on monthly basis [71,72].

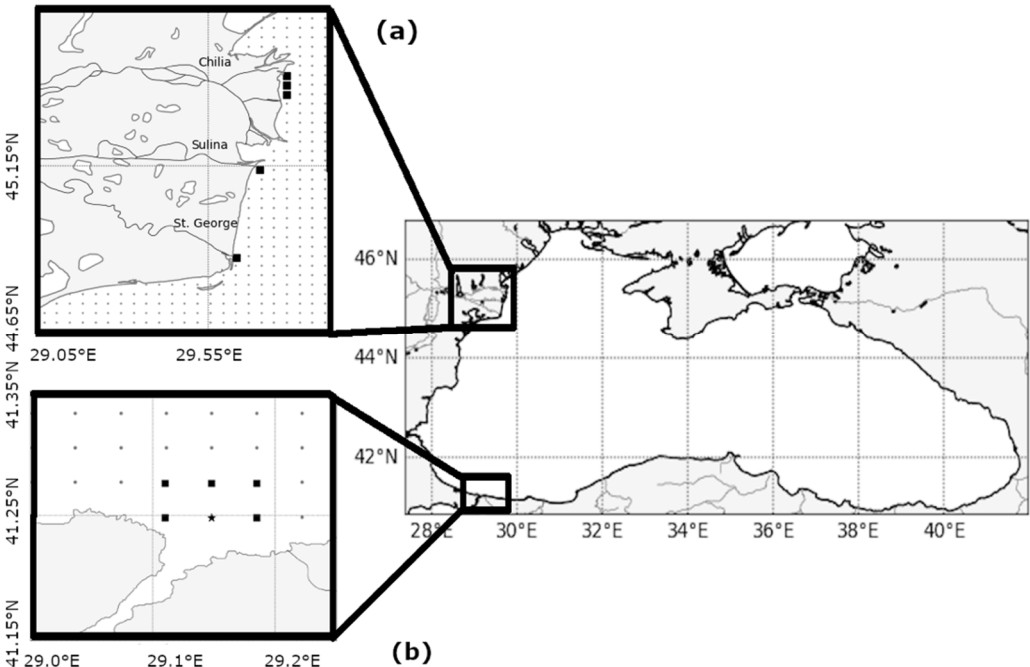

**Figure 2.** (**a**) The Danube Delta: the Chilia, the Sulina and St. George arms are labelled. Small grey circles show the active grid points of the numerical grid in proximity to the Danube delta. Black squares illustrate the Danube arms mouth representation in the hydrodynamical model. (**b**) The Bosphorus Box is used in the hydrodynamical model. Small grey circles show the active grid points of the numerical grid in the Bosphorus area. Black squares illustrate the points in which the temperature and salinity solution were relaxed to [73] profiles. Black star represents the location in which the T and S were relaxed as in the squares, but in addition, the surface boundary condition of "inverse river" is applied.

The representation of saltier and warmer Mediterranean waters that enter the Black Sea is performed by damping the model solution in the area of influence of the Bosphorus exit (Figure 2b) to monthly 3D temperature and salinity climatologies, as computed from [73], at hourly frequencies. The NEMO configuration uses the Turbulent Kinetic Energy vertical mixing scheme for computing the vertical diffusivity and viscosity. Both horizontal eddy viscosity and diffusivity are constant over the domain and set equal to $1.2 \times 10^{-5}$ m$^2$/s

and $1.2 \times 10^{-6}$ m$^2$/s, respectively. The NEMO configuration with main parameters can be found in Zenodo [74].

### 2.3. Wave Model

The wave model used in this work is the third-generation spectral WaveWatchIII [75] version 5.16, hereafter denoted as WW3. The model solves the wave action density balance equation for wavenumber-direction spectra $N(k, \theta, \boldsymbol{x}, t)$, where, $\theta$, is the wave direction, $k$ is the wavenumber, $\boldsymbol{x} = (x, y)$ is the position vector, and $t$ is time. The source terms considered in this implementation concern deep water processes. The wind input source term $S_{in}$ represents the momentum and energy transfer from air to ocean waves, the wave dissipation due to white-capping $S_{ds}$ and the nonlinear transfer by resonant four-wave interactions $S_{nl}$:

$$S = S_{in} + S_{ds} + S_{nl} \tag{1}$$

WW3 has been implemented following WAM Cycle4 model physics [76]. The propagation scheme used is a third-order scheme (Ultimate Quickest) with the "Garden Sprinkler Effect" alleviation method of spatial averaging. Wind input and dissipation are based on [77], in which the wind input parameterization is adapted from Janssen's quasi-linear theory of wind-wave generation [78,79], following adjustments of [80,81]. Nonlinear wave-wave interactions have been modelled using the Discrete Interaction Approximation (DIA) [82,83].

WW3 is a spectral wave model, and therefore requires a discretization of the wave spectra in bins of frequency and direction. The spectral direction over the full cycle is selected and is divided into 24 sectors of 15° width. A first (lowest) frequency of 0.05 Hz was selected, with a frequency increment factor of 1.1. and 30 frequencies in total. The WW3 configuration parameters are provided via Zenodo [74].

### 2.4. Coupling Strategy

In the last few years, several works [7,8,25,27,29,84] deeply described the major wave-currents interaction processes. In this work we consider the wave-currents coupling processes implemented in a recent NEMO release (v4.0) to evaluate the impact of those processes on the Black Sea thermodynamic properties for the first time, with a primary focus on the SST, subsurface temperature and tracer advection due to vertical mixing. To a lesser extent, we gained information on the effect on water velocity, even if the exiguous number of available observations limited this analysis.

The reciprocally forced system between the hydrodynamic model (HM) NEMO and the wind-wave model (WM) WW3 is described in this section. WM and HM models share the same horizontal grid and bathymetry and are forced through a reciprocal hourly field exchange. Eight fields are exchanged between the hydrodynamic and wave models. HM sends to WM the surface currents and the Sea Surface Temperature (SST), which is used to compute the air-sea temperature difference. On the other side, WM provides to HM the following fields: significant wave height (hereafter Hs), mean wave period (Tm), wave peak frequency (Fp), Stokes drift at the surface (SD), wave energy (Φoc) and sea-state dependent momentum flux (τoc). An illustration of the coupling mechanism is given in Figure 3.

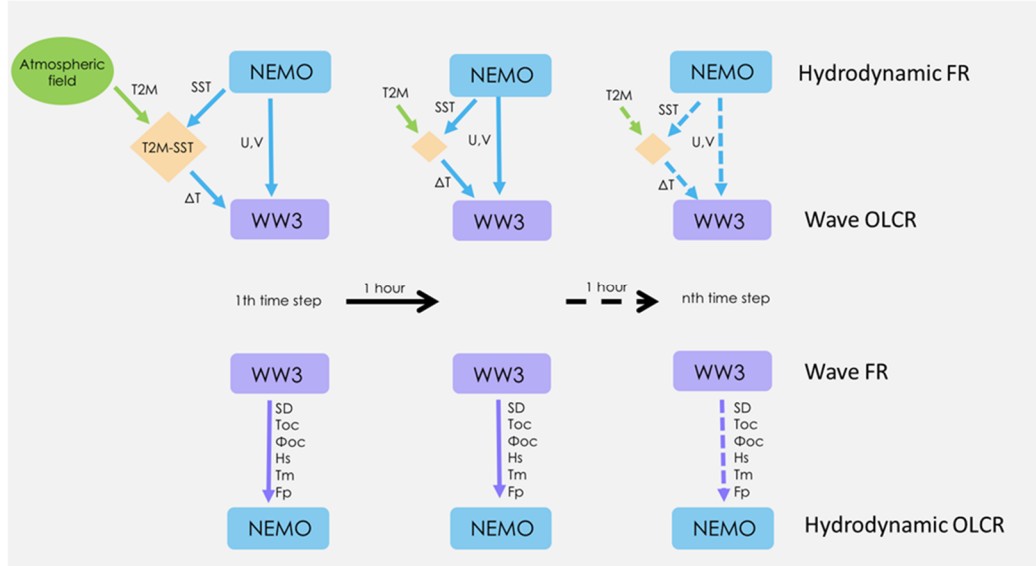

**Figure 3.** Sketch of the forcing mechanism used in this work. Free-runs are marked with FR, forced experiments are marked with OLCR. The fields provide occurs at a one-hour frequency. In the Wave OLCR numerical experiment, NEMO sends current fields (u, v) to WW3 and SST to a system (the light-orange coloured square in the figure), which computes air-sea temperature differences (ΔT) using 2 m temperature from the atmospheric forcing and sends ΔT to WW3. In the hydrodynamic OLCR, WW3 FR sends the Stokes Drift (SD), sea-state dependent momentum flux (τoc), wave energy to ocean energy flux (Φoc), significant wave height (Hs), mean wave period (Tm) and wave frequency peak (Fp) to NEMO.

Five physical processes related to wave-currents interaction are considered: three reflect the impact of waves on hydrodynamics (Figure 4) and two consider the effect of hydrodynamics on the wavefield:

- Sea-state dependent momentum flux;
- Stokes–Coriolis force, which requires a 3D-Stokes drift profile;
- Wave induced turbulence;
- Doppler effect and refraction due to currents;
- Effects of air stability on the growth rate of waves.

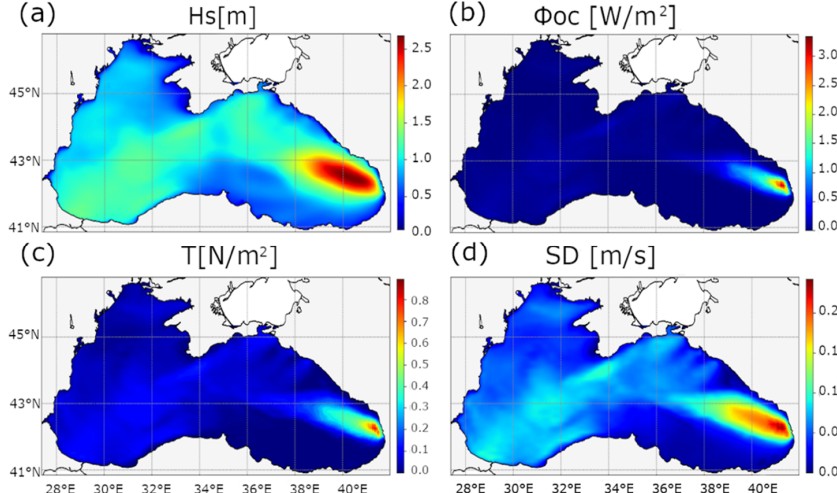

**Figure 4.** Example of wave fields snapshot used to force the hydrodynamic model at a given time step. (**a**) significant wave height, (**b**) wave energy to ocean turbulence, (**c**) sea-state dependent momentum flux and (**d**) Stokes Drift.

### 2.4.1. Sea-State Dependent Momentum Flux

In numerical ocean modelling, the momentum fluxes from the atmosphere to the ocean are traditionally calculated from the wind speed provided by an atmospheric model, using a drag coefficient relating the 10-m winds to the surface stress. With this formulation, the surface fluxes are dependent on the local wind speed only [85]. This formulation does not consider that surface stress is dependent on the sea state, that is, how the wave energy is distributed over the frequency range. Another problem related to this approach is that the net momentum flux is not necessarily conserved between the atmosphere and the ocean. For this reason, several authors continue to investigate the impact of the sea-state on wind stress for more than a half-century [25–27,77,80,86–89].

The air-sea momentum flux, or the total wind stress ($\tau_{tot}$), is the sum of the momentum flux into both surface waves and subsurface currents. A significant proportion of the momentum lost from the atmosphere is taken up by the wind waves as increased wave momentum. The appropriate stress ($\tau_{oc}$) (Equation (2)) to use in an Eulerian ocean model is thus the proportion of the momentum flux that is not taken up by the waves ($\tau_{in}$) plus the momentum lost from the wind-generated waves through dissipation ($\tau_{dis}$) [90].

$$\tau_{oc} = \tau_{tot} - \tau_w \tag{2}$$

$$\tau_w = (\tau_{in} + \tau_{dis}) \tag{3}$$

In a wave model, an important source input is the calculation of the momentum flux from the atmosphere to the ocean, which can be expressed as an integral of the wave variance spectrum multiplied by the wave growth rate (momentum-uptake rate).

$$\tau_w = \rho_w g \int_0^{2\pi} \int_0^{\omega} \frac{k}{\omega} (S_{in} + S_{dis} + S_{nl}) d\omega \, d\Theta \tag{4}$$

where $\rho_w$ is the water density, $k$ is the wavenumber vector, $\omega$ is the wave absolute frequency (in radians), $S_{in}$, $S_{dis}$, $S_{nl}$ are respectively the input, dissipation and non-linear wave model source-term and $\Theta$ is the wave direction.

As in previous work e.g., [24,25,27,57,84] Equation (2), the atmospheric wind stress was corrected as follows

$$\tau_{oc} = \tau_{tot} \left( \frac{\tau_w}{\tau_{tot}} \right) - \tau_w \tag{5}$$

### 2.4.2. Stokes–Coriolis Force and 3D-Stokes Drift Profile

In [9], it is demonstrated that the Stokes drift yields a force on the mean currents. In a rotating ocean, the along-wave crest velocity component of the wave motion is correlated with the vertical component, thus inducing wave stress on the mean Eulerian currents. This stress is proportional to $f \cdot v_s$ where $f$ is the Coriolis force and $v_s$ is the Stokes drift. Following the notation in [91], this term is known as Coriolis–Stokes forcing (Equation (6)).

$$\frac{Du}{Dt} = -\frac{1}{\rho}\Delta p + (u + v_s) \times f \, \hat{Z} + \frac{1}{\rho} \frac{\partial \tau}{\partial Z} \tag{6}$$

where $\hat{Z}$ is the upward vector and $\tau$ is the stress.

Computing the Stokes drift profile is expensive as it involves evaluating an integral with the two-dimensional (2D) wave spectrum at every desired vertical level. It is also often impractical or impossible as the full 2D wave spectrum may not be available. Thus, the full Stokes drift velocity profile is commonly replaced by a monochromatic profile, matched to the transport and the surface Stokes velocity e.g., [86,92–96]. This is problematic, as it is clear that the shear under a broad spectrum is much stronger than that of a monochromatic wave of intermediate wavenumber, due to the presence of short waves, whose associated Stokes drift quickly vanishes with depth. The deep Stokes drift profile will

also be stronger than that of a monochromatic wave, as the low-wavenumber components penetrate much deeper.

Staneva et al. 2017 [25] developed an alternative approximate Stokes drift profile, which has a lower mean-square error deviation than the monochromatic profile for all tested spectra. It has a stronger shear in the upper part and does not tend to zero as rapidly as the monochromatic profile in the deeper part. It mimics the effect of a broader spectrum where the low-wavenumber components penetrate deeper than the mean wavenumber component, while the shorter waves (higher wavenumbers) only affect the upper part of the water column. In this study, the computation of 3D Stokes drift profiles is conducted using the approach in [26], which is included in NEMO v4.0. The Stokes Drift contribution to the water velocity components has been integrated into tracers and momentum equations as in [7,97].

### 2.4.3. Wave Induced Turbulence

Following [11], the effects of breaking waves on upper ocean mixing are explicitly considered by the introduction of the energy flux from waves into the ocean water column. Thus, the transport of turbulent kinetic energy (TKE) must also be introduced. In [11], it is assumed that there is a direct conversion of mechanical energy to turbulent energy at the surface and therefore the turbulent energy flux is assumed to be given by the energy flux from waves to the ocean water column $\Phi$oc (Equation (7)), which follows from the dissipation term in the energy balance equation.

$$\Phi_{\text{oc}} = -\rho_w g \int_0^{2\pi} \int_0^{\infty} S_{ds}\, d\Theta d\omega \tag{7}$$

meaning that the injection of TKE at the surface is given by the dissipation of the wavefield via the sink term in the wave model energy balance equation (usually dominated by wave breaking) and converted into an ocean turbulence source term. In the absence of specific information on the sea state, the energy flux is parametrised in NEMO by considering a typical old wind-sea value. NEMO v4.0 does not include the wave-dependent TKE surface boundary condition, and thus we applied the code modification according to [9].

### 2.4.4. Doppler Effect and Refraction Due to Currents

When surface currents interact with waves, phenomena such as the Doppler shift or refraction occur. The way the Doppler shift modifies the surface waves depends on the current speed relative to the wave propagation speed; thus, slow propagating waves are mainly affected by currents. The effects of currents and waves can merge constructively, creating single exceptionally large waves (rogue waves), or if waves are in opposition to strong currents, they become shorter and steeper (potentially hazardous for navigation) and they can also break. The propagation velocity in the various phase spaces of waves interacting with currents (Equations (8)–(10)) can be written as follows.

$$\dot{x} = c_g + U_c \tag{8}$$

$$\dot{k} = -\left(\frac{\partial}{\partial d}\sigma\right)\left(\frac{\partial}{\partial s}\sigma\right) - K \cdot \frac{\partial}{\partial s}U_c \tag{9}$$

$$\dot{\Theta} = -\frac{1}{k}\left[\left(\frac{\partial}{\partial d}\sigma\right)\left(\frac{\partial}{\partial m}\sigma\right) - K \cdot \frac{\partial}{\partial m}U_c\right] \tag{10}$$

where $c_g$ is the wave propagation velocity vector, $U_c$ is the velocity of the current, $d$ is the water depth, $s$ and $m$ are the directions along and perpendicular, respectively, to the wave direction.

### 2.4.5. Effects of Air Stability on the Growth Rate of Waves

The difference between the sea surface temperature (SST) and the air temperature affects the stability of the lower atmosphere and thus the wind velocity structure. Tolman, 2002 [98] formulated a stability correction by replacing the wind speed with an effective wind speed so that the wave growth reproduces [99] stable and unstable wave growth curves.

The air-sea temperature difference is used to evaluate a stability parameter, $ST$ (Equation (11)), which is written as follows:

$$ST = \frac{hg}{Uh^2} \frac{T_a - T_s}{T_0} \tag{11}$$

where $Uh$ is the wind speed at $h$ height, $T_a$ is air temperature at $h$ height, $T_s$ is the surface temperature and $T_0$ is the reference temperature. $ST$ is used to compute effective wind speed, $U_e$:

$$U_e = U_{10} \left( \frac{c_0}{1 \pm c_1 tanh[c_2(ST - ST_0)]} \right)^{\frac{1}{2}} \tag{12}$$

where $U_{10}$ is the wind speed at 10 m, values $c_0$, $c_1$, $c_2$ and $ST_0$ are set to 1.4, 0.1, 150 and $-0.01$, respectively.

## 3. Numerical Experiments Design and Validation Strategy

Nine numerical experiments were conducted from January 2014 to December 2019, according to the numerical setups explained in the previous section. Five experiments concern hydrodynamics (labelled H) and 4 examine waves (labelled W). All of the numerical experiments are listed in Table 1.

**Table 1.** List of numerical experiments carried out from 2014 to 2019. H refers to hydrodynamics, W refers to waves.

| ID | Experiment | Description |
|----|-----------|-------------|
| H0 | NEMO standalone | NEMO free-run. The hydrodynamic model is a standalone model |
| H1 | NEMO forced via SD | NEMO single field-forced experiment. It uses Stokes Drift at the surface from the WM. Stokes–Coriolis Force (SCF) based on the 3-D reconstruction of the Stokes velocity profile has been computed by the HM |
| H2 | NEMO forced via τoc | NEMO single field-forced experiment. It uses sea-state dependent momentum flux (τoc) from WM |
| H3 | NEMO forced via Φoc | NEMO single field-forced experiment. It uses wave-induced vertical mixing (Φoc) from WM |
| H4 | NEMO forced via SD + τoc + Φoc | NEMO fully-forced experiment. It uses SCF, τoc and Φoc |
| W0 | WW3 standalone | WW3 free-run. The wave model is a standalone model |
| W1 | WW3 forced via $u, v$ | WW3 single field-forced experiment. It uses *currents* from the HM |
| W2 | WW3 forced via ΔT | WW3 single field-forced experiment. It uses ΔT from the HM and atmospheric forcings |
| W3 | WW3 forced via $u, v$ + ΔT | WW3 fully-forced experiment. It uses *currents* and ΔT |

*Validation Strategy and Observational Data*

In this work, the model evaluation was based on GODAE/Oceanpredict [100]. Standard statistics such as BIAS, Root-Mean Squared Error (RMSE), Scatter Index (SI), Pearson's correlation index ($\varrho$) and Standard Deviation (SDev) are used to evaluate the performance of the ocean models by comparing the numerical results against observations (in-situ and/or satellite observed data):

$$BIAS = \frac{1}{n} \sum_{i=1}^{n} (m_i - o_i) \tag{13}$$

$$\text{RMSE} = \sqrt{\frac{1}{n}\sum_{i=1}^{n}(m_i - o_i)^2} \tag{14}$$

$$\text{SI} = \frac{\sqrt{\frac{1}{n-1}\sum_{i=1}^{n}(m_i - o_i - BIAS)^2}}{\underline{o}} \tag{15}$$

$$\rho = \frac{\frac{1}{n-1}\sum_{i=1}^{n}(o_i - \underline{o})(m_i - \underline{m})}{\sqrt{\frac{1}{n-1}\sum_{i=1}^{n}(o_i - \underline{o})^2}\sqrt{\frac{1}{n-1}\sum_{i=1}^{n}(m_i - \underline{m})^2}} \tag{16}$$

$$\text{SDev} = \sqrt{\frac{1}{n}\sum_{i=1}^{n}(x_i - \overline{x})^2} \tag{17}$$

where $o$ and $m$ stand for observed and modelled data, respectively. $x$ applies for both $m$ or $o$; the overbar over a variable denotes the temporal averaged value derived from the sample of length $n$.

Three sources of data, organised by platform, were used to evaluate the accuracy of the model results and described in the following.

- Satellite: including Hs and Sea Surface Temperature (SST). Jason-2 (J2) along-track and quality-controlled altimetric measurements of Hs at 1 Hz sampling frequency (represented in Figure 5a), from the "Archiving, Validation and Interpretation of Satellite Oceanographic data" (AVISO+) have been used for the wave validation.
- SST data are provided by the CMEMS SST Thematic Assembly Center [101]. Night-time L3 satellite data from different space missions are filtered according to quality check, bias-corrected, merged and provided at 1/16° of horizontal resolution. Dataset also provides an error estimate from the optimal interpolation. The operational maintenance of SST data is guaranteed by Consiglio Nazionale delle Ricerche—Istituto di Scienze Marine (CNR ISMAR, Venice, Italy).
- Argo: quality-controlled temperature and salinity in situ vertical profiles used in this work are provided by the CMEMS In Situ TAC [102]. The spatial distribution of almost 1400 Argo floats in the Black Sea basin over the period 2015–2019 is shown in Figure 5b. The operational maintenance of such data is coordinated by the Institute of Oceanology—Bulgarian Academy of Science (IO-BAS, Varna, Bulgaria).
- Moorings: water currents speed and direction data measured from currentometers, shown in Figure 5a, are used to evaluate surface currents from the numerical runs. Data are provided by the CMEMS In Situ TAC [102]. Operational maintenance is guaranteed by the GeoEcoMar Institute (Bucarest, Romania).

Table 2 summarises data type, producers, variables and reference links of the observations used for validating our results.

**Table 2.** List of observation datasets used in this work.

| Dataset | Producer | Variable | Product Name | DOI/URL/Reference |
|---|---|---|---|---|
| SATELLITE | AVISO+ | Hs | Jason-2 Geophysical Data Records (GDR) from precise orbit | https://www.aviso.altimetry.fr/en/data/products/wind/wave-products/gdr-ogdr-osdr-ra2-wwv.html#c6705 (accessed on 1 July 2021) |
| ARGO | CMEMS | T and S vertical profiles | INSITU_GLO_TS_REP_OBSERVATIONS_013_001_b | [102] |
| SATELLITE | CMEMS | SST | SST_BLK_SST_L4_NRT_OBSERVATIONS_010_006 | [101] |
| Mooring EUXRo (01, 02, 03) | CMEMS | Currents speed and direction | INSITU_GLO_TS_REP_OBSERVATIONS_013_001_b | [102] |

Due to the lack of systematic in-situ measurements in the Black Sea (buoys or moorings), wave simulations of Hs are validated by comparing them with the J2 radar altimeter. Satellite Hs measurements [2016–2018] are used to compare the wave model results. Observations for more than 30 min in time or 2 km in geographical space are rejected. Validation was conducted using scatter plots, comparing observed against modelled Hs.

The bivariate probability density function was estimated using a 2D-Gaussian kernel on a squared grid in the variable space provided in [103]. The plots include summary statistics to describe the WW3 skills to evaluate the Hs, such as BIAS, RMSE, scatter index, correlation index, slope and standard error. Salinity (S) and Temperature (T) vertical profiles from the hydrodynamics simulation were validated against ARGO profiles, while SST validation was conducted using L4 satellite data provided in the framework of the Copernicus Marine Environment and Monitoring Service. BIAS and RMSE were computed using numerical results from January 2015 to December 2019. The Hovmöller diagram allows the evaluation of the model capability to reconstruct interannual-annual and seasonal cycles of analysed scalar quantities (e.g., temperature or salinity).

In this work Hovmöller T (S) diagram considered the uppermost 300 m to evaluate the impact of coupling on the temporal variability of water masses formation in the Black Sea, from 2016 to 2019. All the grid points with depth lower than 100 m have been excluded from the computation of the basin mean to avoid contamination from the coastal zones, then, a basin averaged daily mean has been computed.

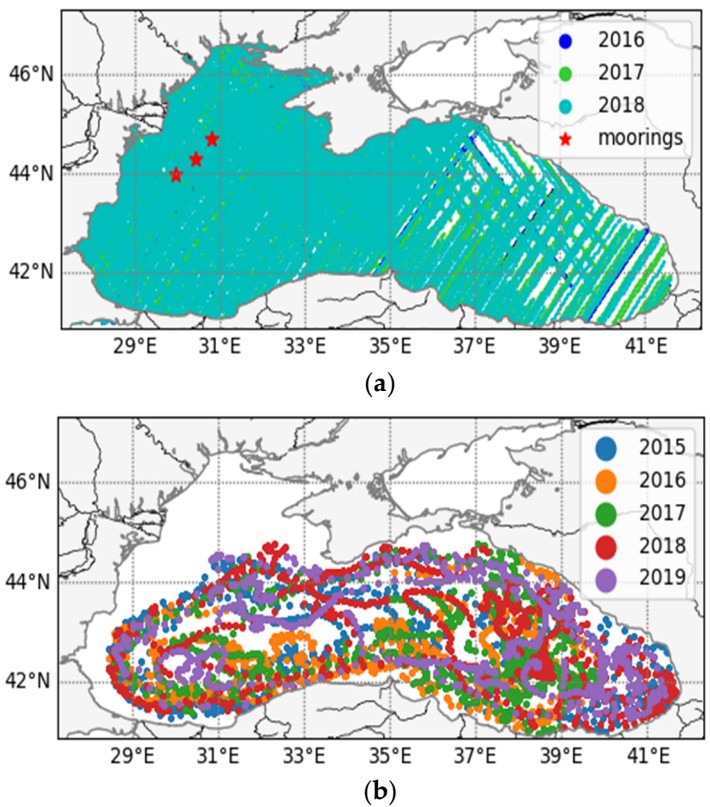

**Figure 5.** Location of the observations used for validating this work. (**a**) Map of Jason-2 satellite tracks from 2016 to 2018 over the Black Sea, and moorings (red stars) location. (**b**) ARGO floats available between 2015 and 2019.

## 4. Results and Discussion

### 4.1. Validation of Hydrodynamical Component

4.1.1. T/S Profiles

Sub-surface temperatures and salinity from HM simulations, as reported in Table 1, were validated using available ARGO profiles from 7.5 m to 1000 m depth as described in Table 2.

Figure 6 shows the relevant metrics for temperature and salinity as given by the set of performed experiments, averaged over the basin for the period 2015–2019. Vertical averaged bias profile for temperature (Figure 6a, left) is negative up to 1000 m and specifically the free-run experiment (H0) is characterised by the largest cold bias in the thermocline layer, up to −0.6 °C, which is only slightly reduced when the Stokes–Coriolis term and wave breaking effects are accounted in experiments H1 and H3 respectively. Temperature bias decreases up to −0.25 °C in the layer 15–40 m when the whole sets of wave-currents interaction processes are considered (H4) and the wave stress (included in experiment H2) provides the largest positive impact in reducing the temperature cold bias in the basin thermocline. In terms of vertical averaged RMSE profile, the envelope of experiments (Figure 6a, mid) shows quite similar values, with higher error in the thermocline, exceeding 2 °C.

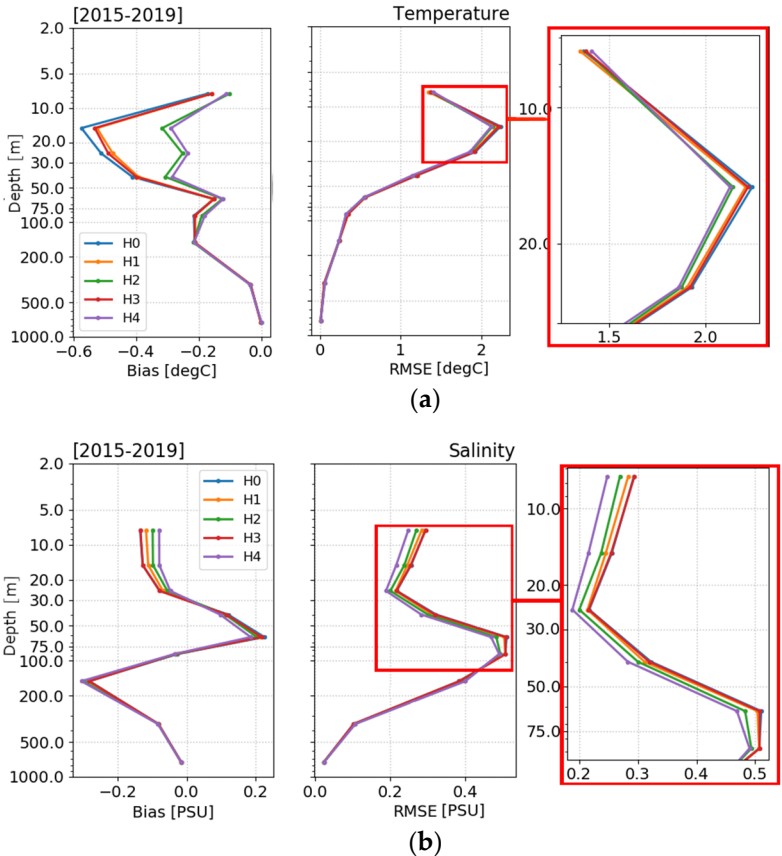

**Figure 6.** Domain averaged 2015–2019 validation of Temperature profiles (**a**) and Salinity (**b**) between 7.5 and 1000 m deep. For each image, the left panel shows BIAS and the right panel shows RMSE. Red boxes represent a zoomed area of the RMSE plots.

The modified ocean stress accounting for the stress absorbed/released by waves is the one providing the best skill among the "single process" experiments. H4 provides the best performance with a halving of BIAS and an RMSE reduction of about 3%, due to enhanced mixing which impacts the mixed layer positioning. Figure 6b reports the metrics for salinity: in particular, the envelope of experiments exhibits quite similar BIAS values,

with a progressive reduction on the BIAS for the fully-forced experiment (H4), dealing with the lowest error at the subsurface (up to 30 m).

Tables 3 and 4 recap skills for the uppermost 200 m at basin scale for both temperature and salinity, respectively: metrics are proposed as averages over every single year and over the period 2015–2019. On a yearly basis, the mean BIAS for temperature (Table 3) is close to zero in H4, but there is no significant difference in terms of RMSE.

**Table 3.** Statistics evaluated by comparing temperature vertical profile measurements and model results from circulation models from 7.5 m to 200 m deep.

| Metric | Experiment | Year 2015 | Year 2016 | Year 2017 | Year 2018 | Year 2019 | Years 2015–2019 |
|---|---|---|---|---|---|---|---|
| | H0 | −0.63 | −0.43 | −0.15 | −0.05 | −0.17 | −0.29 ± 0.075 |
| | H1 | −0.61 | −0.41 | −0.14 | −0.03 | −0.16 | −0.27 ± 0.075 |
| BIAS | H2 | −0.54 | −0.31 | −0.07 | 0.04 | −0.09 | −0.19 ± 0.071 |
| | H3 | −0.61 | −0.41 | −0.13 | −0.05 | −0.17 | −0.27 ± 0.077 |
| | H4 | −0.55 | −0.28 | −0.04 | 0.04 | −0.1 | −0.18 ± 0.073 |
| | H0 | 1.22 | 1.04 | 0.86 | 0.95 | 0.88 | 0.99 ± 0.034 |
| | H1 | 1.2 | 1.02 | 0.86 | 0.94 | 0.89 | 0.98 ± 0.034 |
| RMSE | H2 | 1.19 | 0.91 | 0.88 | 0.98 | 0.89 | 0.97 ± 0.03 |
| | H3 | 1.2 | 1.03 | 0.87 | 0.95 | 0.89 | 0.99 ± 0.032 |
| | H4 | 1.18 | 0.91 | 0.88 | 0.97 | 0.88 | 0.96 ± 0.029 |
| No observations | | 149,928 | 169,312 | 161,799 | 151,030 | 130,869 | 762,938 |

**Table 4.** Statistics evaluated by comparing Salinity vertical profile measurements and model results from circulation models from 7.5 to 200 m deep.

| Metric | Experiment | Year 2015 | Year 2016 | Year 2017 | Year 2018 | Year 2019 | Years 2015–2019 |
|---|---|---|---|---|---|---|---|
| | H0 | −0.09 | −0.15 | −0.06 | 0.08 | −0.03 | −0.05 ± 0.212 |
| | H1 | −0.09 | −0.14 | −0.06 | 0.08 | −0.02 | −0.05 ± 0.208 |
| BIAS | H2 | −0.08 | −0.14 | −0.06 | 0.07 | −0.01 | −0.04 ± 0.207 |
| | H3 | −0.09 | −0.15 | −0.07 | 0.08 | −0.03 | −0.05 ± 0.205 |
| | H4 | −0.09 | −0.14 | −0.08 | 0.08 | −0.01 | −0.04 ± 0.208 |
| | H0 | 0.29 | 0.33 | 0.3 | 0.31 | 0.39 | 0.32 ± 0.128 |
| | H1 | 0.29 | 0.33 | 0.29 | 0.3 | 0.38 | 0.32 ± 0.122 |
| RMSE | H2 | 0.29 | 0.32 | 0.28 | 0.3 | 0.37 | 0.31 ± 0.113 |
| | H3 | 0.29 | 0.33 | 0.3 | 0.31 | 0.38 | 0.32 ± 0.12 |
| | H4 | 0.28 | 0.31 | 0.27 | 0.29 | 0.35 | 0.3 ± 0.115 |
| No observations | | 149,928 | 169,312 | 161,799 | 151,030 | 130,869 | 762,938 |

Table 4 summarises the salinity validation results at the uppermost 200 m. Below 100 m depth, the profiles are very closely aligned. Over the given period, H4 shows a reduction of BIAS of about 20% and RMSE of about 6.5% up to 75 m. We speculate that the average error of 0.3 PSU given by the considered experiments, is related to weak representation of the boundary conditions at the Bosphorus Strait which is not properly representing the Mediterranean waters inflow at all.

The difference between H0 and H4 at the seasonal time scale from 2015 to 2019 was also investigated, as summarised in Table 5, and for the five-year average, the H4 experiment revealed consistently better performance in terms of both salinity and temperature BIAS and RMSE than H0. The best temperature performance in RMSE for the forced run occurs during Winter (Spring) with 0.56 °C (0.61 °C) compared to 0.60 °C (0.66 °C) for H0. During Summer and Autumn, the hydrodynamical model has reduced efficiency (both for H0 and H4 experiments) with almost double RMSE (1.17 °C and 1.18 °C, respectively), and no or negligible differences between H0 and H4.

**Table 5.** Seasonal evaluation of RMSE and BIAS for salinity and temperature for H0 and H4 experiments from 2015 to 2019. Bold indicates the best value between H0 and H4. DJF = Winter, MAM = Spring, JJA = Summer, SON = Autumn.

| Experiment | Metric | Variable | DJF | MAM | JJA | SON |
|---|---|---|---|---|---|---|
| H0 | BIAS | Salinity [PSU] | $-0.06 \pm 0.058$ | $-0.09 \pm 0.116$ | $0 \pm 0.088$ | $-0.03 \pm 0.065$ |
| | | Temperature {°C} | $-0.25 \pm 0.245$ | $-0.33 \pm 0.179$ | $-0.29 \pm 0.243$ | $-0.28 \pm 0.255$ |
| | RMSE | Salinity [PSU] | $0.3 \pm 0.01$ | $0.31 \pm 0.035$ | $0.31 \pm 0.023$ | $0.33 \pm 0.08$ |
| | | Temperature {°C} | $0.66 \pm 0.167$ | $0.60 \pm 0.123$ | $1.17 \pm 0.11$ | $1.23 \pm 0.138$ |
| H4 | BIAS | Salinity [PSU] | $-0.06 \pm 0.56$ | $-0.08 \pm 0.107$ | $-0.01 \pm 0.085$ | $-0.02 \pm 0.06$ |
| | | Temperature {°C} | $-0.19 \pm 0.242$ | $-0.28 \pm 0.176$ | $-0.13 \pm 0.247$ | $-0.16 \pm 0.25$ |
| | RMSE | Salinity [PSU] | $0.29 \pm 0.022$ | $0.29 \pm 0.027$ | $0.29 \pm 0.016$ | $0.30 \pm 0.067$ |
| | | Temperature {°C} | $0.61 \pm 0.162$ | $0.56 \pm 0.104$ | $1.17 \pm 0.143$ | $1.18 \pm 0.099$ |

### 4.1.2. SST

SST validation was conducted using L4 satellite data presents SST BIAS and RMSE for the H4 experiment over the period January 2015–December 2019. Figure 7a shows the geographic distribution of the BIAS and RMSE, Figure 7b displays basin averaged metrics on a monthly timescale. The H4–H0 RMSE difference between fully-forced H4 and free-run H0 is then shown in Figure 7c. The 2D map of computed BIAS shows colder modelled temperature than observations in the North-Western sub-basin (here the absolute minimum BIAS value of about −0.7 °C occurs in the Karkinyts'ka Gulf), in the middle of the Central sub-basin and Central-Western Turkish coasts. Warmer BIAS is shown along the Crimean coasts, in the South-Western and Eastern sub-basin (here the absolute maximum BIAS value of about +0.7 °C occurs off-shore of the area of Trebisonda and Ordu). The RMSE 2D map reveals that most of the basin has an error lower than 0.3 °C, while in the Karkinyts'ka Gulf and along the Eastern coasts the RMSE is greater than 0.5 °C. The RMSE only exceeds 0.7 °C off-shore of Trebisonda and Ordu and on the North-Eastern coast of the Black Sea. So, the Eastern basin exhibits warm waters and high error. The time series of metrics for all considered experiments shown in Figure 7b are characterised by similar BIAS values with interannual and seasonal variability—warm BIAS during summertime due to model overestimation of the measured temperature from satellite—and the lowest error provided by H2 and H4 but still very close to the reference control run H0. The H4-H0 RMSE difference map (Figure 7c) reveals that the wave coupling produces a large improvement in the South-Western area, while in the South-Eastern part the coupling does not reduce the temperature error. The coupling approach also reduces the SST RMSE in the Odessa and Karkinyts'ka gulfs. Table 6 summarises SST BIAS and RMSE averaged at basin scale from 2015 to 2019 for all five hydrodynamic experiments. No significant change is found for BIAS in any of the experiments, or for RMSE in H1 and H3, which have comparable performance to H0 (≈0.88 °C). Performance is only enhanced in H2 and H4 (≈0.85 °C).

As the first conclusion, the considered single- and fully-forced experiments do not significantly ameliorate the skills at basin scale with respect to free-run experiment, apart from some specific regions in the Western basin where the effect of the air-sea interaction combined with waves affects the enhanced mixing and stress determining a general improvement of the model performances. Considering that H4 is our best implementation, all the next validations are mainly showed comparing H0 to H4.

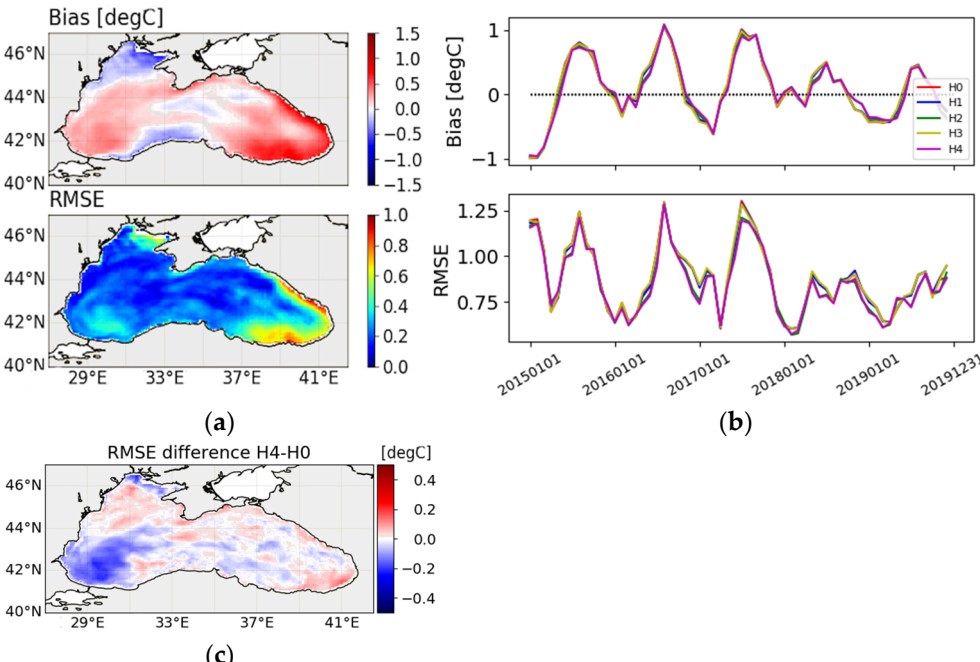

**Figure 7.** 2015–2019 validation of Sea Surface Temperature vs. Satellite observation. (**a**) BIAS (upper panel) and RMSE (bottom panel) for H4 numerical experiment; (**b**) 5-year BIAS (upper panel) and RMSE (lower panel) time series of SST; (**c**) 5-year SST-RMSE difference between H4 and H0 experiments.

**Table 6.** Statistics evaluated by comparing Sea Surface Temperature observations and results from circulation models.

| Metric | Experiment | Years [2015–2019] |
| :---: | :---: | :---: |
| | H0 | 0.10 ± 0.129 |
| | H1 | 0.10 ± 0.128 |
| BIAS | H2 | 0.10 ± 0.127 |
| | H3 | 0.10 ± 0.127 |
| | H4 | 0.09 ± 0.129 |
| | H0 | 0.882 ± 0.088 |
| | H1 | 0.881 ± 0.085 |
| RMSE | H2 | 0.857 ± 0.091 |
| | H3 | 0.883 ± 0.087 |
| | H4 | 0.854 ± 0.09 |

4.1.3. Water Masses

In this section, we present the Hovmöller diagrams for temperature and salinity computed by averaging daily mean fields at the basin scale. The Hovmöller diagram for temperature (Figure 8a) as computed in the free-run experiment H0 well shows that the CIL (around 50–150 m layer) is reducing between 2016 and 2019. This tendency was highlighted in [45], which described time vs. depth basin averaged properties of the Black Sea using available observations from 2005 to 2018. Winter signals, which ventilate the CIL, were weak in 2016, 2018 and 2019, becoming stronger in 2017. This phenomenon is also described in [45], in which CIL is referred to as "perforation". Figure 8b shows instead the temperature difference between H4 and H0 as Hovmöller diagram, highlighting an important mechanism activated by the fully-forced experiment, between 50–100 m, which enhances the CIL reduction as shown in [45] The coupling clearly modified the mixing processes of the basin. During Wintertime, it shows cold and homogeneous temperature differences from the surface up to almost 100 m deep, while starting from Spring, the surface temperature is slightly higher in H4 with a colder temperature core in the subsurface. The

behaviour reaches the maximum at the beginning of Summer, giving a steeper seasonal thermocline for H4. This suggests that when the wave field has stronger activity, the vertical mixing is higher in the forced experiment (Winter), while if the wave activity reduces, the model has lower vertical mixing (Summer). We argued that the main reason for the mixing reduction in the fully-forced experiment might be derived from wave-dependent surface boundary conditions for the TKE. Indeed, while in H4 the amount of TKE at the surface is proportional to wave energy dissipation, in H0 the surface TKE is prescribed using a parameterization for an old wind-sea [17].

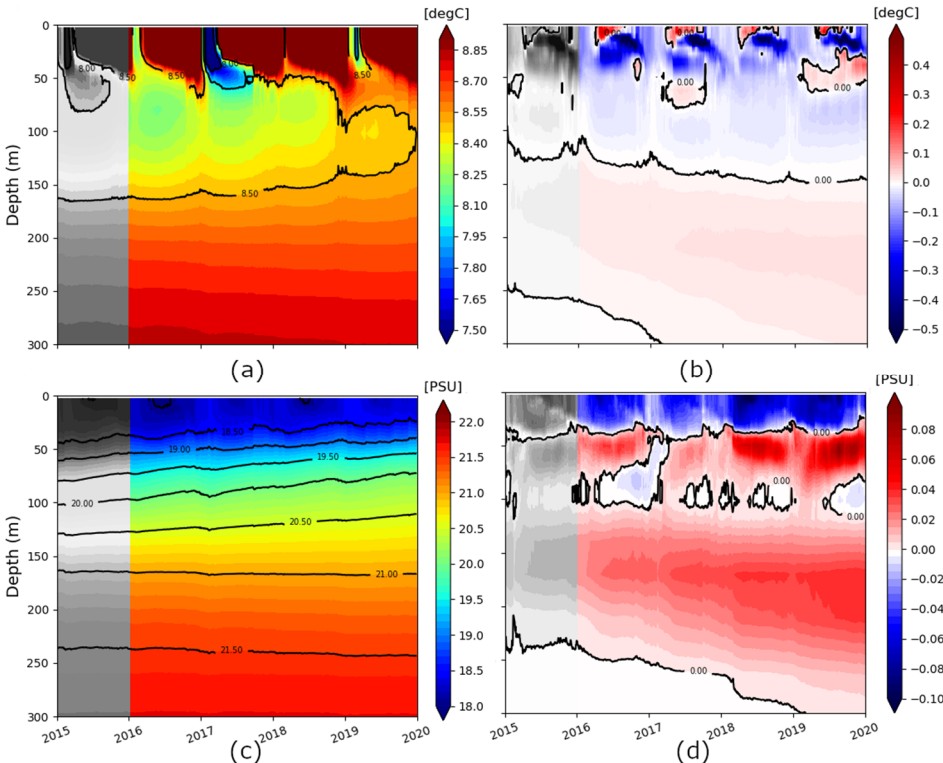

**Figure 8.** Time versus depth versus temperature (**a**,**b**)/salinity (**c**,**d**) diagram for H0 (**a**,**c**) and H4-H0 difference (**b**,**d**). The shaded grey area refers to the model spin-up time. The shaded area refers to the model spin-up time.

The salinity Hovmöller (Figure 8c) from the H0 model results reveals a quite stratified vertical structure. It also reveals increasing salinity in the simulated period, which is not shown in the [45] work, probably due to the closed boundary condition at the Bosphorus. Figure 8d shows the salinity difference between H4 and H0: the fully-forced experiment exhibits slightly fresher waters at the subsurface up to about 50 m and saltier ones up to 100 m. The signal of increased CIL ventilation phenomena in H4, highlighted in the temperature plot (warm water around 50–100 m) is also evident in the salinity diagram (freshwater around 50–100 m).

From Figure 8b,d it is possible to appreciate the difference in the seasonal cycle for salinity and temperature between H4 and H0.

The seasonality of the difference is more evident in 2018. During the first part of the year, warmer-fresher waters from the surface reach almost 40 m deep, while during the second part of the year becomes shallower, reaching the minimum in Autumn. This cycle is easier to observe evaluating temperature (warm core at surface, and cold-core below), while it is more hidden in salinity because the fresher core at the surface is embedded in a whole water column from 0 to 50 m with a negative BIAS. The fact that salinity is always negative in the uppermost 50 m, and saltier below is probably dependent on a higher stratification on average in H4.

### 4.1.4. Currents

Currents speed and direction at 2.5 m depth were validated from 2015 to 2019 using mooring data from CMEMS INS TAC as reported in Table 2 and represented in Figure 5a (red stars). Metrics are reported in Table 7. The main impact due to wave fields on currents statistics is evident when considering the velocity directions, where the coupling reduces the BIAS and RMSE by ~16% and ~10%, respectively, when compared to the control run H0. The improved skill can be mostly attributed to the use of modified stress, showing experiment H2 the lowest BIAS (36°).

**Table 7.** Validation of 2.5 m depth currents speed and direction: averaged statistics using data as provided by the available moorings in the 2015–2019 period.

| Experiment | Variable | BIAS | RMSE |
|:---:|:---:|:---:|:---:|
| H0 | Speed [m/s] | $-0.055 \pm 0.07$ | $0.08 \pm 0.051$ |
| | Direction [°] | $-44 \pm 67$ | $110 \pm 38$ |
| H4 | Speed [m/s] | $-0.054 \pm 0.078$ | $0.08 \pm 0.056$ |
| | Direction [°] | $-38 \pm 66$ | $100 \pm 37$ |

The velocity direction RMSE indicated that the H1, H2 and H3 experiments have comparable errors in the range of 107–111° and that H4 is the best implementation, with a reduced error of around 100°. Conversely to the direction, no significant differences were found for the statistics of currents speed in the forced simulation.

Figure 9 shows the five years averaged currents speed and direction at the mooring locations, and it allows us to appreciate how the currents change when hydrodynamics is forced with waves. Each column in the plot represents a specific mooring: EUXRo01, EUXRo02 and EUXRo03, from left to right.

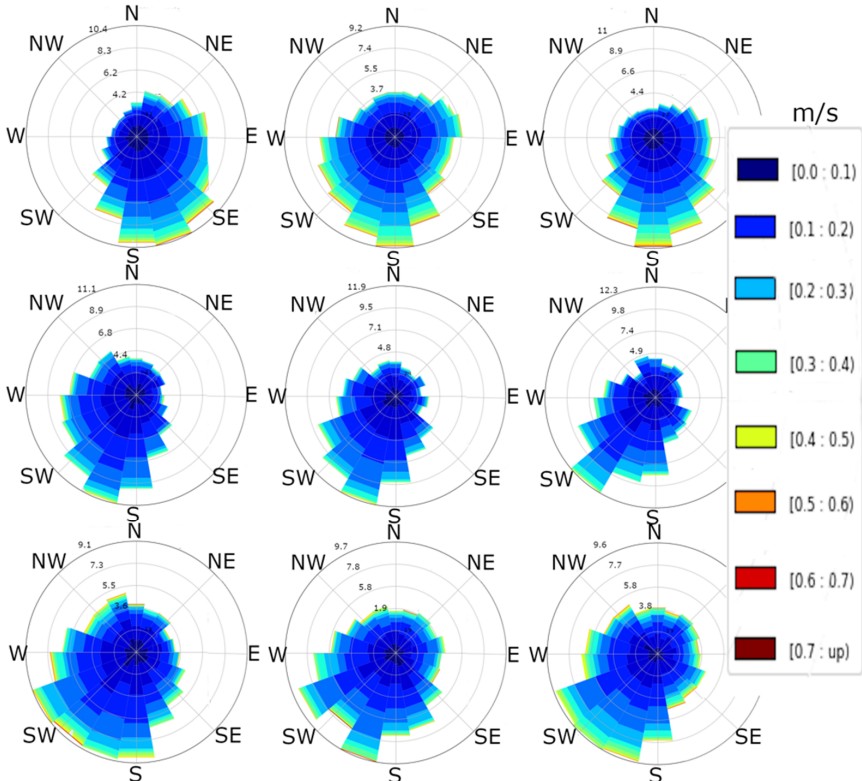

**Figure 9.** Currents speed and direction at 2.5 m deep for moorings (**first row**), H0 (**second row**) and H4 (**third row**). Columns represent the three different mooring locations available.

Additionally, currents speed and direction from observations (first row), from the H0 experiment (second row) and the H4 experiment (third row) is shown. From a qualitative point of view, we can see that in both forced and free-run experiments the model has a stronger western currents component than the observation, probably due to low resolution in space and time of atmospheric forcings, considering also that the observation is taken on a precise location/time.

The main wave direction in this area is aligned with that of the currents (from North-East towards South-West). H0 experiment has a prevalence of one-directional bin, while if waves are considered, the direction of the main current is described by 3–4 directional bins. This explains that the currents simulated by the H4 experiment have a wider dynamic.

Contrarily to H0, in which stronger currents are defined mainly by one direction, the forced experiment H4 has 3–4 directional bins with almost the same speeds and occurrence frequency. Despite this change in magnitude, statistics from Table 5 do not highlight improved skills in the HM simulation. Unfortunately, the very low number of available moorings in the Black Sea region prevents us from performing a more robust validation of the results, but the wider dynamics of H4, which is closer to the observation than H0, seemed to be promising.

### 4.1.5. RMSE vs. Significant Wave Height

To evaluate a correspondence between the improved representation of the ocean physics in the Black Sea thanks to wave-currents forced model, we propose in Figure 10 the time series of daily averaged RMSE differences for temperature (herein, RMSEdif) between fully-forced H4 and free-run H0 experiments in the layer 5–20 m and the corresponding mean, at observation location, of Hs on 2019.

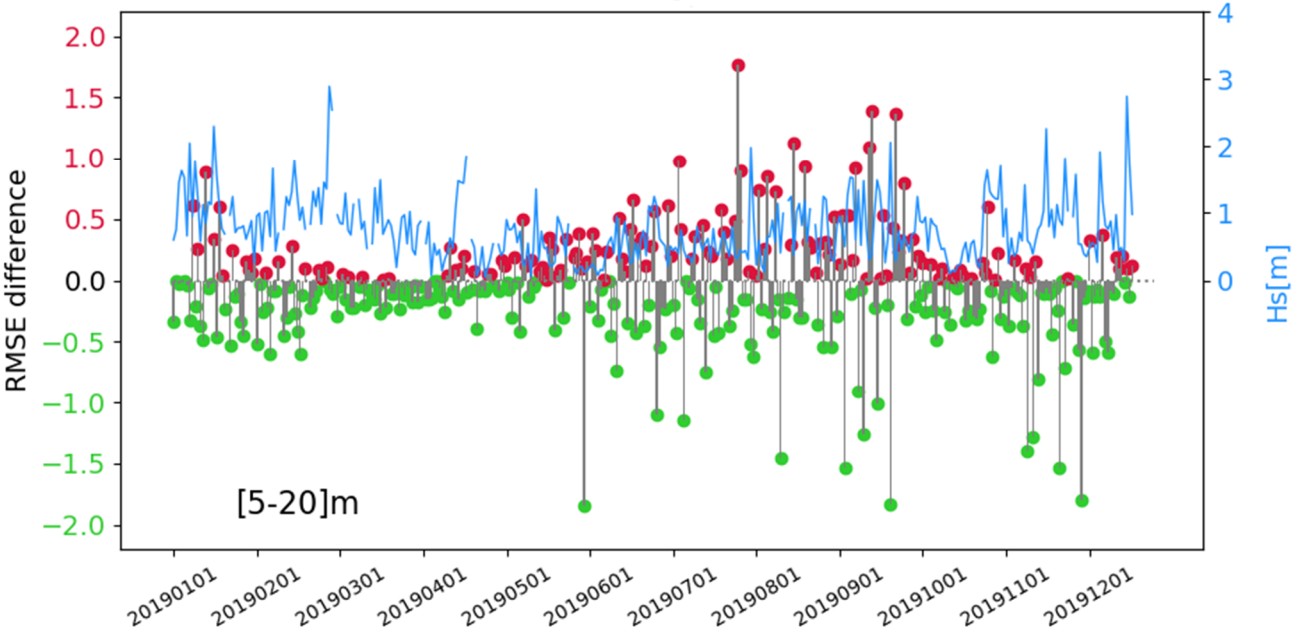

**Figure 10.** Timeseries of daily H4-H0 RMSE difference (RMSEdif) compared to Hs. The grey line represents RMSEdif and is associated with a green or red scatter point if the forced run is respectively better or worse than free-run. The cyan line represents the mean Hs at observation locations.

This plot helps to evaluate whether the positive or negative impact of the coupling is dependent on specific Hs values or the whole Hs spectrum. The figure shows that in general, the H4 has an error lower than H0 in most of the year. The range of variability for RMSEdif evolves seasonally, during Winter and Spring is confined to [−0.5: +1 °C], with the lowest value between mid of February—beginning of April, while during Summer and Autumn the range extends to [−2: +1.8 °C]. From March to May and from September

to October, Hs rarely exceeds 1.5 m and it is always close or below 1.0 m: in this period, the fully-forced run H4 seems to have the best performances (e.g., lower error than H0). This investigation revealed that the forced experiment performs better than the free-run when the sea state has no large fluctuations, as in February–April or in November During Summer and January, in which the variability of the wavefield is high, the forced run still performs better in most of the cases, but there are several days in which the H0 is the best.

As a general indication, the forced model confirms its good performance, demonstrating that in the thermocline region the improvement can be of the order of about 0.5 °C on average. However, to better assess this conclusion, the analysis requires further dedicated investigation over a smaller time scale.

### 4.1.6. Validation for the Wave Component

In this study, a three-year validation (2016–2018) of Hs was conducted using J2 satellite data. The dataset, filtered according to a quality check, consisted of 10,479, 9035 and 10,447 observations for 2016, 2017 and 2018, respectively. Figure 11 compares W0 experiment (a-panel) with W3 experiment (b-panel) considering the whole dataset (29,961 observations).

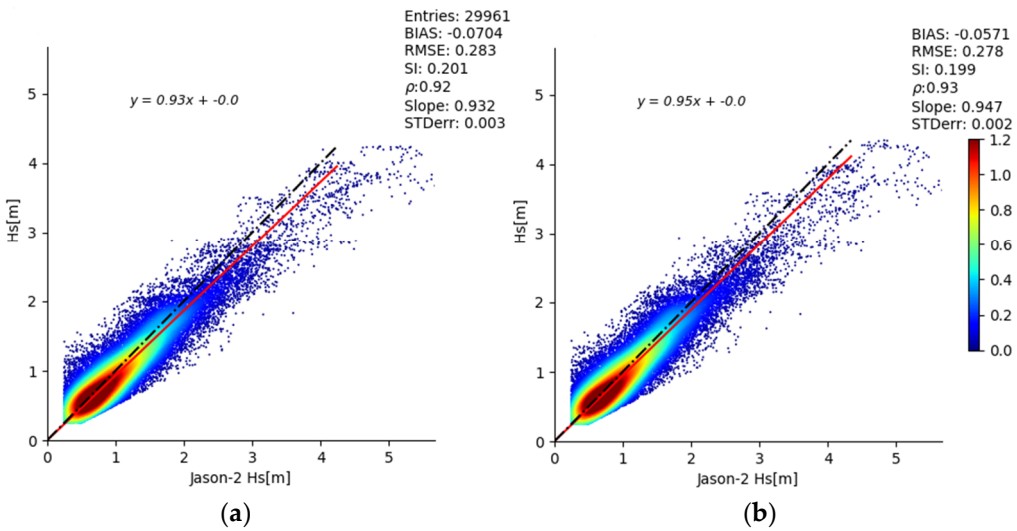

**Figure 11.** Significant wave height validation using Jason-2 satellite from 2016 to 2018. Scatter-plots (**a**,**b**) refer to W0 and W3 experiments, respectively.

The coupling with currents and $\Delta T$ (W3 experiment) induced a performance improvement in all the statistics: BIAS was reduced from $-7$ cm to $-5.7$ cm, RMSE from 28.3 cm to 27.8, the Scatter Index from 0.201 to 0.199, while Pearson's correlation increased from 0.92 to 0.93 and Slope from 0.93 to 0.95.

Table 8 summarises the statistics for all of the wave experiments. All three wave experiments when forced with hydrodynamic fields (W1, W2 and W3) improved model performance, albeit to different extents. The lowest and negligible impact was derived from only-currents (W1 experiment). This result could be a side-effect of the validation method here used. Again, the absence of observations, e.g., from buoy wave gauge in this case, strongly affected our capability to validate the experiments and we were obliged to use satellite data, which has two main disadvantages: it is not reliable near the coast, where currents are stronger and may impact the waves; it does not provide information about wave direction, which could be affected by refraction phenomena.

On the contrary, $\Delta T$ (W2 experiment), which acts mainly on Hs has been positively evaluated and confirmed what is demonstrated in the literature [29]. When both currents and $\Delta T$ were considered (W3 experiment), the lowest error was obtained, with a reduction of $\approx -18\%$ in BIAS, $\approx -2\%$ in RMSE. Even the precision of the simulation has been improved, with reduction of scatter index and increasing 1% in $\rho$ in correlation.

**Table 8.** Significant wave height (m) validation statistics.

| Metric | Experiment | Year 2016 | Year 2017 | Year 2018 | Years 2016–2018 |
|--------|-----------|-----------|-----------|-----------|-----------------|
| BIAS | W0 | −0.077 | −0.071 | −0.06 | −0.070 ± 0.007 |
| | W1 | −0.075 | −0.070 | −0.062 | −0.069 ± 0.005 |
| | W2 | −0.062 | −0.062 | −0.054 | −0.058 ± 0.004 |
| | W3 | −0.59 | −0.059 | −0.053 | −0.057 ± 0.003 |
| RMSE | W0 | 0.304 | 0.274 | 0.270 | 0.283 ± 0.015 |
| | W1 | 0.302 | 0.271 | 0.269 | 0.282 ± 0.015 |
| | W2 | 0.299 | 0.270 | 0.266 | 0.279 ± 0.015 |
| | W3 | 0.297 | 0.267 | 0.266 | 0.278 ± 0.014 |
| SI | W0 | 0.216 | 0.194 | 0.190 | 0.201 ± 0.011 |
| | W1 | 0.215 | 0.192 | 0.191 | 0.200 ± 0.011 |
| | W2 | 0.215 | 0.192 | 0.190 | 0.200 ± 0.011 |
| | W3 | 0.214 | 0.190 | 0.190 | 0.199 ± 0.011 |
| slope | W0 | 0.893 | 0.977 | 0.934 | 0.932 ± 0.034 |
| | W1 | 0.895 | 0.978 | 0.933 | 0.933 ± 0.034 |
| | W2 | 0.908 | 0.993 | 0.946 | 0.946 ± 0.035 |
| | W3 | 0.91 | 0.994 | 0.946 | 0.974 ± 0.034 |
| No observations | | 10,479 | 9035 | 10,447 | 29,961 |

## 5. Conclusions

In this study, we investigated the importance of wave-currents interaction in the Black Sea for the first time. A reciprocally forced numerical system has been implemented using the ocean circulation model NEMO v4.0, which is now including the most important wave-currents physics, and the third-generation wave model WaveWatchIII. The coupling consists of providing Sea Surface Temperature and surface currents to the wave model, computed by the hydrodynamic model and returning sea-state dependent momentum flux, Surface Stokes Drift and wave dissipated energy to ocean vertical mixing. Even if our main focus was the assessment of tracers, a positive effect has been found also on waves.

The inclusion of wave-currents interaction in the Black Sea hydrodynamics, determined reduction of the RMSE for SST ($\approx -3.5\%$) and the upper ocean from 7.5 to 200 m water depths ($\approx -3\%$). The main differences between the forced and free runs are related to the uppermost part of the water column (depth < 35 m). The strongest impact on the vertical profile caused by the for is related to the sea-state dependent momentum flux, while the Stokes–Coriolis force and the sea-state dependent vertical mixing have negligible effects. On average, the coupling produced greater benefits in Winter and Spring, which are characterised by intense wave activity and low vertical stratification. According to our validation, the forced run was found to perform better than the free run for moderate wave heights. In general, the forced run demonstrated a slightly warmer water temperature than the free one.

The forced experiment had a positive impact even on salinity, with a reduction of $\approx -10\%$ in BIAS and $\approx -6.5\%$ in RMSE on the uppermost averaged 200 m.

The main impact of waves on currents concerned the reduction of direction BIAS, without improvement/worsening for the velocity module. Anyway, we noticed a wider dynamic in direction and speed for the currents field in the fully-forced experiment.

We inferred that the physical process which has been improved in forced hydrodynamics is related to the vertical mixing, with larger mixing during Winter and lower mixing during Spring–Summer with respect to free-run experiments, as showed in Section 4.1.3.

Coupling also improved the wave model performance, which slightly better represents the Hs satellite observations, in forced configuration. The results indicate that the improvement was mainly related to the better representation of the effect of air-sea temperature differences on the wave growth, while the usage of the surface currents plays

a minor role, as already shown in [28]. Unfortunately, the lack of wave buoys in the basin prevented the analysis of the coupling impact on wave period and direction.

In conclusion, this preliminary coupling configuration produced a modest but clear improvement in the simulations of temperature, salinity, currents and waves in the Black Sea. Future works could investigate this setup-up over a longer time scale to evaluate its impact on climatological time scale as in [19,104,105]. The use of an external coupler to conduct online field exchange is an important technical development that can be considered in further studies.

**Author Contributions:** S.C. designed the forced system and coordinated the work in collaboration with S.A.C., G.C. and P.L., E.C. participated in the scientific discussion sharing the theoretical approach as implemented in the Mediterranean Sea. All authors have read and agreed to the published version of the manuscript.

**Funding:** This research was funded by the Copernicus Marine Environment and Monitoring Service for the Black Sea Monitoring and Forecasting Centre, contract n. 72.

**Institutional Review Board Statement:** Not applicable.

**Informed Consent Statement:** Not applicable.

**Conflicts of Interest:** The authors declare no conflict of interest.

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
