# Peer review of "A Modelling Approach for the Assessment of Wave-Currents Interaction in the Black Sea"

_jmse, doi:10.3390/jmse9080893_

Round 1

Reviewer 1 Report

Review of “Wave-currents interaction in the Black Sea: evaluation of a new modelling approach for the next generation of operational forecasting systems” by Salvatore Causio, Stefania A. Ciliberti, Emanuela Clementi, Giovanni Coppini, Piero Lionello

The manuscript aims at describing and evaluating the skills of a new ocean-wave modelling framework for the Black Sea that will be used for the next generation operational forecasting systems of this area. The analysis is conducted computing BIAS and RMSE of ocean and wave hindcasts simulations against various type of observations for the period 2015-2019.

The topic is quite interesting for the broad scientific community since to the best of my knowledge this is the first reported ocean-wave coupled system of the Black Sea area. However, I think that the manuscript is still at a very early stage of development and needs significant major improvements before being suitable for publication. Therefore, I recommend rejection of the present form of this paper.

The following are the reasons behind my suggestion:

  1. The authors claim that one of the main novelties of the study is that they have implemented a two-way off-line coupled system. This is quite an “against the tide” choice, since the mainstream solution (see e.g. Clementi et al. 2017, Lewis et al. 2019a,b, Couvelard et al. 2020 ) is to use an external coupler to efficiently exchange on-line fields between model components. However, the authors do not describe what are the benefits or drawbacks of their alternative approach: is it more accurate? is it faster? Why this approach was preferred with respect to the mainstream one?
  2. The abstract is quite “catchy” but do not fully reflect the results reported later in the manuscript (see conclusions of the manuscript).
  3. In my opinion the introduction is quite unfocused and do not provide a comprehensive review of
    1. The state-of-the-art of
      1. coupling techniques and coupled modelling systems
      2. ocean and wave modelling approaches in the Black Sea
    2. The main physical mechanisms driving the Black Sea circulation
    3. The typical wind and wave regime of the Black Sea
    4. The main aspects of the Black Sea hydrodynamic that could be improved by considering ocean-wave interactions
  4. The modelling system description is quite unfocused (e.g., what is the rationale behind figure 2, why the way Danube’s and Bosporus’ fluxes are parameterised in the ocean model is so important for a coupled system?) and doesn’t give all the necessary information (e.g., a section describing in detail the coupling strategy).
  5. The methods include a quite general technical description of the 5 wave-currents interactions implemented in this study. However, what I think would be really interesting to know more about but is not mentioned at all is why the authors decided to implement these specific wave-current interactions? Which physical processes were they aiming to improve?
  6. The description of the observational datasets used for models’ skill assessment is quite general; not very informative - I think more details should be given, especially for future comparisons and results reproducibility.
  7. When describing the metrics used to evaluate model performance, no information about how uncertainty affecting statistics is measured or at least included in the evaluation is given.
  8. When presenting the results
    1. Only a comparison of metrics is given, with very little scientific insights or perspective.
    2. Skills of runs switching on only one wave-interaction per time are mostly ignored and only the differences between the uncoupled and the fully coupled runs are discussed (usually with no scientific detail or investigation as in point a.)
    3. No information about the uncertainty associated with computed metrics is given.
    4. To me it seems that improvements for BIAS seems quite important, while for RMSE are very small (or negligible). I think this deserves more attention, since it could depend on compensation errors affecting the BIAS metric.
    5. Consider spatially and long-term averaged metrics may not be the best approach to evaluate the impact of ocean-wave coupling, especially for a future forecasting system which will be employed for short time-scales predictions.
    6. I can not see the rationale behind section 4.1.3: there is no impact of ocean-wave coupling on CIL dynamics. Also, how is the Hovmoller diagram obtained? Where the authors expecting any effect on the CIL? If yes, why? Also, what is the CIL? I think this important feature of the Black Sea circulation was never introduced (see point 3.).
    7. I think section 4.1.4 is very confusing: so, wave-current interactions affect only current direction or also magnitude? Then, can the authors comment on the possible reasons behind that?
    8. I don’t understand why there is a section 4.1.5. In my opinion these results should be presented in section 4.1.1. Also, can the authors give some information on the uncertainty associated with their results?
    9. In my opinion the text in section 4.1.6 do not describe entirely what is shown in figure 20. Also, as the authors report, the largest improvements with the coupled system are obtained for moderate waves, which in the case of the Black Sea wave regime would correspond to an average Hs of 0.5 m.  However, wave-current interactions are likely to represent a leading order process during extreme events characterised by large waves (e.g. Staneva et al. 2016, Wu et al. 2019). Can the author comment on this?
    10. What is the rationale behind section 4.1.7: I cannot see the differences for CW that the authors are referring to. Also, I would expect that the main impact would be in shallow areas. Can the authors explain then why the wave model is tailored for deep water dynamics?
    11. I think the validation of the wave component suffers of many of the above points.
  9. In general, the text and the figures could be improved

References:

Staneva, J., Wahle, K., Koch, W., Behrens, A., Fenoglio-Marc, L., & Stanev, E. V. (2016). Coastal flooding: Impact of waves on storm surge during extremes - A case study for the German Bight. Natural Hazards and Earth System Sciences, 16(11), 2373–2389. https://doi.org/10.5194/nhess-16-2373-2016

Clementi, E., Oddo, P., Drudi, M., Pinardi, N., Korres, G., & Grandi, A. (2017). Coupling hydrodynamic and wave models: First step and sensitivity experiments in the Mediterranean Sea. Ocean Dynamics, 67(10), 1293–1312. https://doi.org/10.1007/s10236-017-1087-7

Lewis, H. W., Castillo Sanchez, J. M., Arnold, A., Fallmann, J., Saulter, A., Graham, J., et al. (2019a). The UKC3 regional coupled environmental prediction system. Geoscientific Model Development, 12(6), 2357–2400. https://doi.org/10.5194/gmd-12-2357-2019

Lewis, H. W., Castillo Sanchez, J. M., Siddorn, J., King, R. R., Tonani, M., Saulter, A., et al. (2019b). Can wave coupling improve operational regional ocean forecasts for the north-west European Shelf?. Ocean Science, 15(3), 669–690. https://doi.org/10.5194/os-15-669-2019

Wu, L., Staneva, J., Breivik, Ø., Rutgersson, A., Nurser, A. G., Clementi, E., & Madec, G. (2019). Wave effects on coastal upwelling and water level. Ocean Modelling, 140, 101405. https://doi.org/10.1016/j.ocemod.2019.101405

Couvelard, X., Lemarie, F., Samson, G., Redelsperger, J.-L., Ardhuin, F., Benshila, R., Madec, G. (2020). Development of a two-way-coupled ocean-wave model: assessment on a global NEMO(v3.6)-WW3(v6.02) coupled configuration, Geoscientific Model Development, 13(7), 2020, 3067-3090, https://doi.org/10.5194/gmd-13-3067-2020

Author Response

Review of “Wave-currents interaction in the Black Sea: evaluation of a new modelling approach for the next generation of operational forecasting systems” by Salvatore Causio, Stefania A. Ciliberti, Emanuela Clementi, Giovanni Coppini, Piero Lionello

The manuscript aims at describing and evaluating the skills of a new ocean-wave modelling framework for the Black Sea that will be used for the next generation operational forecasting systems of this area. The analysis is conducted computing BIAS and RMSE of ocean and wave hindcasts simulations against various type of observations for the period 2015-2019.

The topic is quite interesting for the broad scientific community since to the best of my knowledge this is the first reported ocean-wave coupled system of the Black Sea area.

However, I think that the manuscript is still at a very early stage of development and needs significant major improvements before being suitable for publication. Therefore, I recommend rejection of the present form of this paper.

The following are the reasons behind my suggestion:

  1. The authors claim that one of the main novelties of the study is that they have implemented a two-way off-line coupled system. This is quite an “against the tide” choice, since the mainstream solution (see e.g. Clementi et al. 2017, Lewis et al. 2019a,b, Couvelard et al. 2020 ) is to use an external coupler to efficiently exchange on-line fields between model components. However, the authors do not describe what are the benefits or drawbacks of their alternative approach: is it more accurate? is it faster? Why this approach was preferred with respect to the mainstream one?

This work aimed to investigate the effects of a set of wave-current coupling processes in the Black Sea. In this first step, we decided to use the off-line coupling  that allowed us to analyze how and at which extent an offline coupling with a wave model could affect the main physical fields in the basin to evaluate the new physical ocean-wave parameterization within NEMO v4.0.  We believe this is the first step to build the knowledge in the Black Sea basin towards online coupling approaches for operational forecasting system implementation. Considering that we obtained a small but still significant improvement, we are prone to move towards an online coupling.

  1. The abstract is quite “catchy” but do not fully reflect the results reported later in the manuscript (see conclusions of the manuscript).

We thank the reviewer for general comments that helped hopefully to improve the overall paper. Following the revisions made to address the comments, we revised the abstract and a new version is now proposed.

  1. In my opinion the introduction is quite unfocused and do not provide a comprehensive review of
    1. The state-of-the-art of
      1. coupling techniques and coupled modelling systems
      2. ocean and wave modelling approaches in the Black Sea
    2. The main physical mechanisms driving the Black Sea circulation
    3. The typical wind and wave regime of the Black Sea
    4. The main aspects of the Black Sea hydrodynamic that could be improved by considering ocean-wave interactions

We propose a revision of the introduction to better address reviewer’s comments. We detailed the importance of the wave-currents coupling and the main modelling approaches. Then we selected some examples of strategies pursued in regional systems - Baltic, North Sea, Mediterranean and North West European Shelf - to finally introduce our objective that is to develop a coupled wave-currents model in the Black Sea. We described the main coupling techniques, giving references to previous works. The section has been enriched with more details about the oceanography in the basin and its wave-wind climate. Review of the main ocean and wave modelling approaches have been reported, too.

  1. The modelling system description is quite unfocused (e.g., what is the rationale behind figure 2, why the way Danube’s and Bosporus’ fluxes are parameterised in the ocean model is so important for a coupled system?) and doesn’t give all the necessary information (e.g., a section describing in detail the coupling strategy).

We understand this point and we tried to restructure the section as well, keeping consistency in terms of providing details for the hydrodynamical model as for the wave model description. The section on coupling deserves a more detailed description, in our opinion, being the core of the paper. In describing the Danube and the Bosporus peculiarities the aim was really to support a proper overall description of the hydrodynamical model setup, including current limits that will be overcome in the next evolution of the physical core. They are in any case not strictly related to the coupling itself, but we believe that an overall general description would serve the scope. We provided the full details about the NEMO settings in Appendix A and about WW3 in Appendix B, with the scope to share as much as possible the numerical choices on the basis of our setup.

  1. The methods include a quite general technical description of the 5 wave-currents interactions implemented in this study. However, what I think would be really interesting to know more about but is not mentioned at all is why the authors decided to implement these specific wave-current interactions? Which physical processes were they aiming to improve?

In section 2.3 “Coupling strategy” we reshaped the overall description of the proposed and implemented coupling strategy by introducing the scheme and then the processes we aimed to solve.

  1. The description of the observational datasets used for models’ skill assessment is quite general; not very informative - I think more details should be given, especially for future comparisons and results reproducibility.

In section 3.1 “Validation strategy and observational data” we added subsections to describe the observational datasets as shown in Wu et al., 2019.

  1. When describing the metrics used to evaluate model performance, no information about how uncertainty affecting statistics is measured or at least included in the evaluation is given.

In this study, we proposed mainly the average accuracy of best-estimate fields from a 4-year simulation for temperature, salinity, sea surface temperature and currents. These statistics indicate the overall quality of a product based on first and second order Gaussian statistics. This is described in the Hernandez et al. 2015 and represents a standard in Copernicus Marine Service. We understand that the uncertainty assessment is relevant: if the proposed analysis is still considered insufficient, despite this clarification, we need to integrate the evaluation, but it would require more than 1 week for the provisioning of the feedbacks. We will look forward to having specific indications on how this revision will evolve.

  1. When presenting the results
    1. Only a comparison of metrics is given, with very little scientific insights or perspective.

The overall Section 4 has been revised in order to include a more focused explanation of numerical results. Some portions of the subsections have been largely modified.

    1. Skills of runs switching on only one wave-interaction per time are mostly ignored and only the differences between the uncoupled and the fully coupled runs are discussed (usually with no scientific detail or investigation as in point a.)

We tried to embed the suggestion in the related section, with comments on different single-field experiments. We stressed the most important processes and then we underlined that because the H4 represented the best implementation, we focused on this in the next validations.

    1. No information about the uncertainty associated with computed metrics is given.

For this please refer to comment on point 7.

    1. To me it seems that improvements for BIAS seems quite important, while for RMSE are very small (or negligible). I think this deserves more attention, since it could depend on compensation errors affecting the BIAS metric.

All the discussion about the performance skills have been revised, avoid to put attention on BIAS but giving more importance to stronger statistics as RMSE

    1. Consider spatially and long-term averaged metrics may not be the best approach to evaluate the impact of ocean-wave coupling, especially for a future forecasting system which will be employed for short time-scales predictions.

We agree on this and of course we will keep into consideration this comment when we will setup the coupled version for operational forecasting. Thanks to the reviewer  comments, we realized that the scope of the work was not properly focused and considering the type of simulations that span over 4 years we set an evaluation on the overall quality of numerical results.

    1. I can not see the rationale behind section 4.1.3: there is no impact of ocean-wave coupling on CIL dynamics. Also, how is the Hovmoller diagram obtained? Where the authors expecting any effect on the CIL? If yes, why? Also, what is the CIL? I think this important feature of the Black Sea circulation was never introduced (see point 3.).

We understand the comment and for this reason we would propose an overall revision of Section 4.1.3. Following the reviewer's suggestions, we propose a plot on the difference between coupled and uncoupled experiments to comment on the impact of the coupling on the vertical structure of temperature and salinity at basin scale. Of course, the expected impact is at subsurface thanks to induced mixing from waves. We also introduced the CIL in the Black Sea general description (in the introduction, being a fundamental water mass property in the basin) and we detailed how the Hovmoller diagrams have been obtained. We would like to maintain the diagrams because they highlight the difference in temporal formation of water masses between coupled and uncoupled experiments.

    1. I think section 4.1.4 is very confusing: so, wave-current interactions affect only current direction or also magnitude? Then, can the authors comment on the possible reasons behind that?

We revised the overall Section 4.1.4 providing an explanation of the plots and some conclusions. However, the limited number of observations prevent us to provide a more detailed investigation at basin scale on the impact of the wave-currents coupling on currents. The main outcome from this initial analysis confirmed that the coupled experiment is able to produce more dynamics than the uncoupled one and much closer to the observation.

    1. I don’t understand why there is a section 4.1.5. In my opinion these results should be presented in section 4.1.1. Also, can the authors give some information on the uncertainty associated with their results?

We understand the comment and for this reason we would propose to delete Section 4.1.5  merging it with section 4.1.1. Please refer to point 7 for what concerns the uncertainties.

    1. In my opinion the text in section 4.1.6 do not describe entirely what is shown in figure 20. Also, as the authors report, the largest improvements with the coupled system are obtained for moderate waves, which in the case of the Black Sea wave regime would correspond to an average Hs of 0.5 m.  However, wave-current interactions are likely to represent a leading order process during extreme events characterised by large waves (e.g. Staneva et al. 2016, Wu et al. 2019). Can the author comment on this?

The section has been totally revised and we propose a more focused description of Figure 10 (which we guess is the figure you are referring to). Regarding the second part of the comment we believe that an analysis of extreme events is extremely interesting but for the scientific baseline of the paper was considered an action to further develop (and propose maybe for a future work).

    1. What is the rationale behind section 4.1.7: I cannot see the differences for CW that the authors are referring to. Also, I would expect that the main impact would be in shallow areas. Can the authors explain then why the wave model is tailored for deep water dynamics?

We understand the comment and for this reason we would propose to delete Section 4.1.7 since water mass properties as introduced here are out of scope considering the general objective of the work.

    1. I think the validation of the wave component suffers of many of the above points.

We tried to capitalize the comments the reviewer provided for the previous sections to also improve the wave one. However, we stressed in the text and in the introduction that our major aim was the understanding of the waves' impact on hydrodynamics, nevertheless for completeness we also provided some basic explanation with validation exercises to assess the impact of hydrodynamics on waves.

  1. In general, the text and the figures could be improved.

We thank the reviewer for the comments: we tried to improve the text keeping in mind as much as possible the overall list of feedbacks. Regarding the figures, we modified some figures but  we really would like to have more indications on which ones need further improvements: we tried to make them sufficiently readable and we hope now the comments help to interpret them.

Reviewer 2 Report

The manuscript entitled “Wave-currents interaction in the Black Sea: evaluation of a new modelling approach for the next generation of operational forecasting systems” by Salvatore Causio, Stefania A. Ciliberti, Emanuela Clementi, Giovanni Coppini, and Piero Lionello investigated a coupled ocean-wave model performance in the Black Sea by considering the wave-current interactions. The manuscript is well written, and findings can be interesting. However, extensive improvements are necessary before considering acceptance.

Major comments:

  1. The authors carried out several long-term numerical experiments considering the 5 wave-current related processes and showed a great amount of model validations, however, there lacks sufficient in-depth analysis regarding how and why the coupled model outperformed the stand-alone cases. For example, the authors highlighted the importance of sea-state dependent momentum flux in reducing the general model errors but did not show any details regarding under what sea state conditions did the wave coupling outperforms. Did the wave coupling enhance the model performance by amending the water-side wind stress? I would recommend selecting a representative period when the coupling effects are significant, such as Jan. 2017 and showing the difference of simulated sea states between H0 and H4 (e.g., wave parameters, surface roughness and shear stress). In addition, Table 3 and 4 show the validations of water temperature and salinity in each year, however, there is no corresponding analysis regarding how and why the model (coupled and uncoupled) behave differently in each year. In the end, the authors concluded that “The coupled run was found to perform better than uncoupled run for moderate wave heights”. Is this general conclusion or it is specifically for the Black Sea? I would expect intuitively that the coupled model might outperform under a rougher sea state.
  2. Section 2.3.1 is very cursory. The air-sea momentum flux and its dependence on sea states (the most influential process indicated in the manuscript) should be clearly described. Please refer to Charnock (1955), Janssen (1989, 1993), Taylor and Yelland (2001), Oost (2002), Breivik (2015, 2016), Staneva (2017) etc.

Specific Comments:

  1. Figure 8 and Figure 9. I would suggest including subplots with difference between H0 and H4 for both water temperature and salinity. Please justify the statement” No major differences were found between the vertical structure of the coupled and uncoupled runs, with only a slight warmer behavior in the coupled run.” As shown in Table 3, bias reduced from -0.29 in the uncoupled run to -0.18 in the coupled run.
  2. The modeled water temperature and salinity show great match with observations while the simulated currents seem to have big variance. Did the simulations include any data assimilation or nudging of water temperature and salinity?
  3. Line 512. Figure indexes and statements should be consistent, the same in Line 525, 536, 559 and throughout the manuscript.
  4. Line 559 Figure 11. What does the colormap represent? Please revise.
  5. Table 7. I guess the bias in 2016 is a typo. Please correct.
  6. Section 4.2 looks cursory, again, there are just statistic numbers. I would expect to include analysis in terms of how waves in your coupled model applications are affected by currents.

Author Response

The manuscript entitled “Wave-currents interaction in the Black Sea: evaluation of a new modelling approach for the next generation of operational forecasting systems” by Salvatore Causio, Stefania A. Ciliberti, Emanuela Clementi, Giovanni Coppini, and Piero Lionello investigated a coupled ocean-wave model performance in the Black Sea by considering the wave-current interactions. The manuscript is well written, and findings can be interesting. However, extensive improvements are necessary before considering acceptance.

Major comments:

  1. The authors carried out several long-term numerical experiments considering the 5 wave-current related processes and showed a great amount of model validations, however, there lacks sufficient in-depth analysis regarding how and why the coupled model outperformed the stand-alone cases. For example, the authors highlighted the importance of sea-state dependent momentum flux in reducing the general model errors but did not show any details regarding under what sea state conditions did the wave coupling outperforms. Did the wave coupling enhance the model performance by amending the water-side wind stress? I would recommend selecting a representative period when the coupling effects are significant, such as Jan. 2017 and showing the difference of simulated sea states between H0 and H4 (e.g., wave parameters, surface roughness and shear stress). In addition, Table 3 and 4 show the validations of water temperature and salinity in each year, however, there is no corresponding analysis regarding how and why the model (coupled and uncoupled) behave differently in each year. In the end, the authors concluded that “The coupled run was found to perform better than uncoupled run for moderate wave heights”. Is this general conclusion or it is specifically for the Black Sea? I would expect intuitively that the coupled model might outperform under a rougher sea state.

We thank the reviewer for the important comment. We revised the text to better explain what is our aim in this paper, which was not so clear in the first version. We were more interested in the evaluation of the general impact of coupling wave-hydrodynamics than the evaluation in a smaller time frame as already described in literature. From our thought, this way to evaluate the coupling represents a novelty, because it is common to find in literature evaluation of coupling in a specific time period. In our idea, the evaluation of the coupling on a smaller timescale, mainly related to extreme events, will be evaluated in a next and dedicated paper, in which we would like to apply an on-line coupling. However, if the reviewer is interested in this, we could include the analysis on a small time scale, but we need more than 1 week.

The fact that the reviewer noticed that the models have different performance in different years is exactly the reason why we would like to maintain an overall valuation based on 5 years. We could have selected a year or a period to confirm that "coupling is better"... or "coupling is worse". We found both these 2 conditions in our analysis but we were more interested in an averaged evaluation to understand if really the coupling could be worth or not.

As the reviewer underlined, in general the coupled runs outperform under a rougher sea state, and we confirmed this assumption considering the seasonal average, in which in Winter we had the best performance and higher Significant wave height on averaged basin scale. Anyway, we can' t infer from our results and figure 11 that higher is the wave, higher is the improvement. In addiction, as now highlighted also in section 4.1.3 for vertical mixing, we think that the coupling here applied acts" on the whole wave spectra, giving improvement, even if at a lesser extent, also with calm sea-state.

  1. Section 2.3.1 is very cursory. The air-sea momentum flux and its dependence on sea states (the most influential process indicated in the manuscript) should be clearly described. Please refer to Charnock (1955), Janssen (1989, 1993), Taylor and Yelland (2001), Oost (2002), Breivik (2015, 2016), Staneva (2017) etc.

We thank the reviewer for suggestion and we added more details to the section

Specific Comments:

  1. Figure 8 and Figure 9. I would suggest including subplots with difference between H0 and H4 for both water temperature and salinity. Please justify the statement” No major differences were found between the vertical structure of the coupled and uncoupled runs, with only a slight warmer behavior in the coupled run.” As shown in Table 3, bias reduced from -0.29 in the uncoupled run to -0.18 in the coupled run.

We completely remodeled the section and the figure in the new version to answer a previous comment of reviewer 1.

  1. The modeled water temperature and salinity show great match with observations while the simulated currents seem to have big variance. Did the simulations include any data assimilation or nudging of water temperature and salinity?

No, in this case the runs are performed without data assimilation for Temperature and Salinity. We believe this can be determined by the air-sea interaction and the kind of validation: we are using hourly means, however the moorings measure at fixed time and location.

  1. Line 512. Figure indexes and statements should be consistent, the same in Line 525, 536, 559 and throughout the manuscript.

Thanks for the comment, we fixed it

  1. Line 559 Figure 11. What does the colormap represent? Please revise.

We used colors just to distinguish if RMSEdif is positive or negative. We added the description in the figure caption. Do the reviewer prefer we describe colors also in the text?

  1. Table 7. I guess the bias in 2016 is a typo. Please correct.

Thanks for the comment, we fixed it.

  1. Section 4.2 looks cursory, again, there are just statistic numbers. I would expect to include analysis in terms of how waves in your coupled model applications are affected by currents.

We revised the whole paper focusing mainly on the impact of waves in hydrodynamic properties of the basin. However we would like to provide information about the impact of hydrodynamics on waves even though this represents a marginal topic.

Reviewer 3 Report

This study proposes a novel implementation of wave-current interactions in the Black Sea for future operational forecasting systems, which consists of a coupled two-way off-line numerical system based on the ocean circulation model NEMO v4.0 and the third-generation wave model WaveWatchIII v5.16. The results are within the scope of the JMSE. I suggest a major revision.

Comments:

  1. Some previous studies related to the present study should be considered in the Introduction section, e.g., Hsiao, et al., 2020. On the Sensitivity of Typhoon Wave Simulations to Tidal Elevation and Current. J. Mar. Sci. Eng., 8(9), 731; Hsiao, et al., 2019. Quantifying the contribution of nonlinear interactions to storm tide simulations during a super typhoon event. Ocean Engineering, 194, 106661.
  2. The fully coupled wave-circulation modeling systems have been developed for maybe two decades, what is the novelty in your modeling system? More details should be provided in Discuss or Conclusions section.

Author Response

This study proposes a novel implementation of wave-current interactions in the Black Sea for future operational forecasting systems, which consists of a coupled two-way off-line numerical system based on the ocean circulation model NEMO v4.0 and the third-generation wave model WaveWatchIII v5.16. The results are within the scope of the JMSE. I suggest a major revision.

Comments:

  1. Some previous studies related to the present study should be considered in the Introduction section, e.g., Hsiao, et al., 2020. On the Sensitivity of Typhoon Wave Simulations to Tidal Elevation and Current. J. Mar. Sci. Eng., 8(9), 731; Hsiao, et al., 2019. Quantifying the contribution of nonlinear interactions to storm tide simulations during a super typhoon event. Ocean Engineering, 194, 106661.

We thank the reviewer for the comment and we included the dissertation of the suggested work in the introduction, being part of the state-of-the-art.

  1. The fully coupled wave-circulation modeling systems have been developed for maybe two decades, what is the novelty in your modeling system? More details should be provided in Discuss or Conclusions section.

We tried to address better the focus of this work during this first round of revisions: in our knowledge, the Black Sea has been not sufficiently studied from this perspective, despite the huge literature for the global ocean and regional configuration. In the perspective of improving the capacity of reconstructing the past state of the Black Sea as well as for forecasting capacities, we believe that starting a more focused analysis on the effect of waves on hydrodynamics is fundamental. We propose a new drafted ending sections to defend this position and scope.

Round 2

Reviewer 1 Report

Second review of “Wave-currents interaction in the Black Sea: evaluation of a new modelling approach for the next generation of operational forecasting systems” by Salvatore Causio, Stefania A. Ciliberti, Emanuela Clementi, Giovanni Coppini, Piero Lionello

First of all, I would like to thank the authors for their effort in trying to address my previous comments in such a short time - in my opinion the result is an improved manuscript with respect to the previous version. However, I also think that the paper still needs major revisions before being suitable for publication in JMSE.

The following are my major comments:

  1. I think the new proposed title still misses the main aim of the paper: it still focuses on the modelling approach, which I think is not the main novelty of this study and neither reflects the real aim of the paper. In addition, the word ‘understanding’ is not quite appropriate in my opinion since the paper is not really trying to investigate the physical mechanisms underpinning the impact of wave-current interactions in the Black Sea. I suggest: ‘Assessing wave-current interactions in the Black Sea’ as a better title.
  2. In the second revised version the authors try to clarify their ‘off-line coupling’ modelling approach, also providing two references for it. However, I think the term ‘off-line coupling’ is still ambiguous and can potentially induce misunderstanding. Typically, when information from an external system is applied as a boundary forcing (in this case, via file IO) with no feedbacks between the systems the terminology used is “forced simulation” while the definition of “coupled simulation” refers usually to a modelling system where the various model components interactively exchange fields, often via an external on-line coupler (Breivik et al. 2015, Law Chune et al. 2018, the same references given by the authors). In order to be consistent with other works in this field, I suggest the authors clarify that this work is on the impact of “wave forcing” and “ocean forcing” on ocean and wave models of the Black Sea, respectively.
  3. I think the abstract and the introduction could be significantly improved. In particular, I find much of the new text on the Black Sea circulation not relevant for the topic of this paper. I think the authors should provide some information and relevant references for the main characteristics of the Black Sea circulation, focusing on the ones that will be investigated in the study. Also, there are no references to previous studies using NEMO in the Black Sea and the authors might want to add the following recent studies:
    • Bruciaferri et al. 2020. The development of a 3D computational mesh to improve the representation of dynamic processes: The Black Sea test case. https://doi.org/10.1016/j.ocemod.2019.101534  
    • Ehsan Sadighrad et al. 2021. Mesoscale eddies in the Black Sea: Characteristics and kinematic properties in a high-resolution ocean model. https://doi.org/10.1016/j.jmarsys.2021.103613
  4. I think that the authors could easily include (e.g.) the standard deviation associated with their averages to give more perspective to their results.
  5. I had serious difficulties in reviewing the section on SST: text is missing or meaningless, figures are overlapping or duplicated, what is EAN? … I strongly suggest to improve the text and layout of this section.
  6. Regarding the water masses, I think the plots are now clearer and suggest some interesting physical mechanisms. However, I don’t think the explanation of the authors completely reflects what is happening. The authors state that “The coupling clearly modified the mixing processes of the basin. During Wintertime, it shows cold and homogeneous temperature difference from surface up to almost 100m deep, while starting from Spring, the surface temperature is slightly higher in H4 with a colder core temperature in the subsurface”. I think what the figure shows is that the major differences for temperature are in summer, with a stronger seasonal pycnocline (arguably thermocline) in the H4 run which reduces the mixing with respect to H0. However, can the authors explain why there is less mixing in H4 than in H0 during summer? Also, this seasonal effect on the vertical mixing does not seem to affect salinity, which presents a consistent cold bias between H4 and H0 during the whole year. Could the authors explain the reason behind those differences affecting the vertical mixing of two active tracers?
  7. If the authors will decide to address these comments, I strongly suggest that the new revised paper should be in a much cleaner and easy-to-review format than this second one, which I found extremely hard to review given the not consistent track of the changes, new text in different colours, text missing, old and new figures overlapping etc. …

The following are additional minor comments:

  1. Please add the first name of the Author of a paper when a reference is in the beginning of a sentence – this will greatly help in improving readability.
  2. Sometimes you use “wave-current interactions” while others “wave-currents interaction” … please chose one form and try to be consistent.
  3. The authors might be interested in adding the following references for recent studies on the impact of ocean-wave coupling:
    • Staneva et al. 2021. Effects of wave-induced processes in a coupled wave–ocean model on particle transport simulations. https://doi.org/10.3390/w13040415
    • Bruciaferri et al. 2021. The impact of ocean-wave coupling on the upper ocean circulation during storm events. https://doi.org/10.1029/2021JC017343
  4. My previous comment on the Hovmoller diagram was referring in what data are you using / plotting. From your explanation I can only guess they are basin averages with time frequency of days? … please clarify.
  5. I am not an English native speaker. However, while I think the previous version was written in a quite fluent and clear English, I find this second version less correct, regarding both style and grammar. I recommend checking the English language.
  6. I think the format of text and tables could be improved.
  7. Please check the references format – in this version there are now additional numbers - e.g. 28. 3 Clementi, E., Oddo, P., Drudi, M., Pinardi, N., Korres, G., Grandi, A. Coupling hydrodynamic and wave models: first step and sensitivity experiments in the Mediterranean Sea. Ocean Dyn. 2017, 67.10, pp. 1293–1312
  8. While I strongly support the sharing of the model namelist to reproduce model results, the information given in appendix, as it is now, is not very useful and sometimes incorrect (e.g., “no sleep” condition does not exist). I suggest the authors could share the complete NEMO namelist as external resources, for example via open dissemination research data repository (e.g. zenodo) which will also associate a doi to their data.

References:

Breivik, Ø., Mogensen, K., Bidlot, J.R., Balmaseda, M., Janssen, P A. E. M. Surface wave effects in the NEMO ocean model: Forced and coupled experiments: Waves in NEMO. en. In: J. Geophys. Res. Oceans 2015, 120.4, pp. 2973–2992. doi: 10.1002/2014JC010565

Law Chune, S. and Aouf, L.: Wave effects in global ocean modelling: parametrizations vs. forcing from a wave model, Ocean Dynam.,2018, 68, 1739–1758, https://doi.org/10.1007/s10236-018-1220- 2, 2018.

Reviewer 3 Report

The authors have well addressed my comments, I recommend that this paper can be accepted for publication in the JMSE.

Author Response

We thanks the reviewer for the suggestions.

Round 3

Reviewer 1 Report

I thank the authors for answering all of my comments, and I am happy that they addressed most of them. I just have few minor comments after which I am happy to recommend the paper for publication in JMSE.

Minor comments:

  1. I would recommend to carefully check the English language - e.g. I think that, in the case of the title, the correct form is either 'A modelling approach for assessing ...' or 'A modelling approach for the assessment ...'
  2. Please when you cite an author at the beginning of a sentence add the complete reference - e.g. L53 Janssen et al. 2013, I think this will improve readability.
  3. I think the authors should consider adding the reference to 
    • Bruciaferri et al. 2020. The development of a 3D computational mesh to improve the representation of dynamic processes: The Black Sea test case. https://doi.org/10.1016/j.ocemod.2019.101534

when mentioning previous studies focusing on modelling mesoscale dynamics and water mass formation in the Black Sea (L103-104). In my opinion this study is particularly relevant for this paper: it analyses the skills of CMEMS Black Sea reanalysis which were produced with a model very similar to the one used by the authors in the present study, describing similar bias and rmse to the ones reported by the authors in Fig.6 of the manuscript and discussing possible reasons behind them.

  1. If you decide to share the complete NEMO namelist via an open dissemination research data repository (e.g. zenodo), please provide in the text the reference to the doi associated with your data repository.

Author Response

  1. I would recommend to carefully check the English language - e.g. I think that, in the case of the title, the correct form is either 'A modelling approach for assessing ...' or 'A modelling approach for the assessment ...'

Thanks to the reviewer for the comment. We checked the whole text.

  1. Please when you cite an author at the beginning of a sentence add the complete reference - e.g. L53 Janssen et al. 2013, I think this will improve readability.

Thanks for clarification, we applied the change

  1. I think the authors should consider adding the reference to
  • Bruciaferri et al. 2020. The development of a 3D computational mesh to improve the representation of dynamic processes: The Black Sea test case. https://doi.org/10.1016/j.ocemod.2019.101534

when mentioning previous studies focusing on modelling mesoscale dynamics and water mass formation in the Black Sea (L103-104). In my opinion this study is particularly relevant for this paper: it analyses the skills of CMEMS Black Sea reanalysis which were produced with a model very similar to the one used by the authors in the present study, describing similar bias and rmse to the ones reported by the authors in Fig.6 of the manuscript and discussing possible reasons behind them.

Thanks to the reviewer for the comment. We integrated the suggested reference.

  1. If you decide to share the complete NEMO namelist via an open dissemination research data repository (e.g. zenodo), please provide in the text the reference to the doi associated with your data repository.

We shared the namelists via Zenodo and we put the reference (to the DOI) as suggested by the reviewer